# Stress-induced host membrane remodeling protects from infection by non-motile bacterial pathogens

Caroline Tawk[1,2] (iD), Giulia Nigro[3], Ines Rodrigues Lopes[4,5], Carmen Aguilar[1], Clivia Lisowski[1], Miguel Mano[4] (iD), Philippe Sansonetti[3] (iD), Jörg Vogel[2,6] (iD) & Ana Eulalio[1,5,*] (iD)

## Abstract

While mucosal inflammation is a major source of stress during enteropathogen infection, it remains to be fully elucidated how the host benefits from this environment to clear the pathogen. Here, we show that host stress induced by different stimuli mimicking inflammatory conditions strongly reduces the binding of *Shigella flexneri* to epithelial cells. Mechanistically, stress activates acid sphingomyelinase leading to host membrane remodeling. Consequently, knockdown or pharmacological inhibition of the acid sphingomyelinase blunts the stress-dependent inhibition of *Shigella* binding to host cells. Interestingly, stress caused by intracellular *Shigella* replication also results in remodeling of the host cell membrane, *in vitro* and *in vivo*, which precludes re-infection by this and other non-motile pathogens. In contrast, *Salmonella* Typhimurium overcomes the shortage of permissive entry sites by gathering effectively at the remaining platforms through its flagellar motility. Overall, our findings reveal host membrane remodeling as a novel stress-responsive cell-autonomous defense mechanism that protects epithelial cells from infection by non-motile bacterial pathogens.

**Keywords** acid sphingomyelinase; host stress response; membrane remodeling; *Salmonella*; *Shigella*

**Subject Categories** Membrane & Intracellular Transport; Microbiology, Virology & Host Pathogen Interaction

The EMBO Journal (2018) 37: e98529

## Introduction

Intestinal epithelial cells (IECs) constitute a physical and biochemical barrier between host and microorganisms. While not considered professional immune cells, IECs are crucial contributors to immune surveillance and the initial inflammatory responses against infection (Sansonetti, 2004; Peterson & Artis, 2014). For example, IECs secrete pro-inflammatory cytokines and chemokines in response to infection by certain enteroinvasive bacterial pathogens such as *Shigella flexneri* and *Salmonella* Typhimurium. This, in turn, leads to a massive infiltration of professional immune cells into the sites of inflammation, from which ensues a local increase in reactive oxygen species and a profound hypoxia (Colgan & Taylor, 2010; Zeitouni *et al*, 2016; Arena *et al*, 2017). Such change in environment not only constrains the pathogen but also dramatically affects the intestinal epithelium. Interestingly, while this inflammation-related stress used to be considered as posing additional harm to the affected tissues, there is recent evidence to suggest that it makes crucial contributions to counteract infection processes at the cellular level (Chovatiya & Medzhitov, 2014).

Bacterial pathogens have evolved sophisticated mechanisms to subvert and often invade the intestinal epithelium, thereby overcoming host defense mechanisms (Sansonetti, 2004; Pizarro-Cerda & Cossart, 2006; Carayol & Tran Van Nhieu, 2013). *Shigella flexneri*, for example, crosses the intestinal barrier by transcytosis through M-cells (Wassef *et al*, 1989; Perdomo *et al*, 1994), before invading IECs from the basolateral side (Mounier *et al*, 1992). For invasion, *Shigella* uses its type III secretion system (T3SS) to inject effector proteins into target cells to subvert host defense pathways, promoting its own internalization by a trigger mechanism that involves the formation of actin-rich membrane ruffles (Ogawa *et al*, 2008; Schroeder & Hilbi, 2008; Parsot, 2009).

Notwithstanding the diverse active mechanisms used by bacteria to mediate binding and invasion of host cells [e.g., pili, fimbriae, adhesins/invasins, T3SS (Pizarro-Cerda & Cossart, 2006; Stones & Krachler, 2016)], it has also become clear that efficient bacterial entry requires permissive sites in the host membrane. Membrane rafts, which are highly dynamic membrane domains enriched in sphingolipids and cholesterol that mediate the compartmentalization of signaling proteins and receptors (Lingwood & Simons, 2010; Sezgin *et al*, 2017), have been shown to be utilized by numerous

1  Host RNA Metabolism Group, Institute for Molecular Infection Biology (IMIB), University of Würzburg, Würzburg, Germany
2  RNA Biology Group, Institute for Molecular Infection Biology (IMIB), University of Würzburg, Würzburg, Germany
3  Molecular Microbial Pathogenesis Laboratory, Institut Pasteur, Paris, France
4  Functional Genomics and RNA-based Therapeutics, UC-BIOTECH, Center for Neuroscience and Cell Biology (CNC), University of Coimbra, Coimbra, Portugal
5  RNA & Infection Group, UC-BIOTECH, Center for Neuroscience and Cell Biology (CNC), University of Coimbra, Coimbra, Portugal
6  Helmholtz Institute for RNA-Based Infection Research (HIRI), Würzburg, Germany
   *Corresponding author. Tel: +351 231249170; E-mail: ana.eulalio@uni-wuerzburg.de; aeulalio@ci.uc.pt
   [The copyright line of this article was changed on 3 December 2018 after original online publication.]

bacterial pathogens (reviewed in Refs: Lafont & van der Goot, 2005; Bagam *et al*, 2017). For example, *Shigella* uses its IpaB effector protein to bind the host raft-associated CD44 transmembrane receptor (Lafont *et al*, 2002); entry of *Listeria monocytogenes* into host cells requires the localization of the host receptors E-cadherin and HGF-R/Met in specific lipid domains (Seveau *et al*, 2004). In addition to receptors, plasma membrane composition itself, specifically cholesterol and sphingolipid membrane content, impacts the binding and internalization of various bacterial pathogens, including *Shigella* and *Salmonella* species (Garner *et al*, 2002; Lafont *et al*, 2002; Misselwitz *et al*, 2011a; Santos *et al*, 2013).

Here, we investigated the impact of the general stress response of epithelial cells on the infection by *S. flexneri* and *Salmonella* Typhimurium. We found that induction of stress in epithelial cells by inflammatory cues and oxidative insults prevents the binding of *Shigella*, a non-motile pathogen, to host cells. We demonstrate that this inhibition results from extensive remodeling of the host plasma membrane following a stress-induced activation of the acid sphingomyelinase (ASM). By contrast, the related motile pathogen *Salmonella* can overcome this barrier, using flagellar motility to reach and accumulate at the remaining permissive entry sites. Moreover, we show that intracellular replication of *Shigella* activates ASM and subsequent membrane remodeling, thus suppressing re-infection by non-motile pathogens. Collectively, our findings demonstrate a role for the host stress response in protecting cells against *Shigella* infection and demonstrate the involvement of ASM and membrane remodeling in this process.

# Results

## Host cell response to stress inhibits *Shigella* infection

To investigate whether host cell stress has a deleterious effect on the outcome of *Shigella* infection, we treated HeLa cells, an epithelial cell line commonly used to study *Shigella* infection, with sub-lethal concentrations of sodium arsenite (Fig 1A). Arsenite is widely used to induce oxidative stress (Bernstam & Nriagu, 2000; Liu *et al*, 2001). Following arsenite removal, cells were extensively washed and then infected with *Shigella;* infection efficiency was monitored at early, intermediate, and late stages of infection (0.5, 2, and 6 hpi, respectively; Fig 1A) by: (i) fluorescence microscopy, (ii) colony-forming unit (cfu) assays, and (iii) qRT–PCR. Interestingly, pre-treatment of cells with arsenite strongly reduced *Shigella* infection, at all time points tested (4.7- to 8.8-fold compared to control, cfu; Figs 1B and D, and EV1A and B). Validating these observations, *Shigella* infection was also inhibited by arsenite in all tested colon epithelial cells, namely HCT-8, HT-29, and Caco-2 cells (Figs 1C and D, and EV1B–D).

Response to environmental stress in eukaryotic cells generally dampens bulk protein synthesis due to impaired mRNA translation initiation (Holcik & Sonenberg, 2005). However, inhibition of translation by puromycin and cycloheximide did not affect *Shigella* infection (Fig EV1E–G), demonstrating that the effect of arsenite on *Shigella* infection is unrelated to translation shutdown.

To understand whether inhibition of *Shigella* infection by cellular stress is a broad phenomenon, we tested other stress inducers, namely anisomycin, hydrogen peroxide ($H_2O_2$), hypoxia, or the inflammatory cytokine TNF-$\alpha$. All the stress inducers have been widely used to mimic conditions encountered by cells during inflammation (arsenite, anisomycin) or are *per se* stimuli present during inflammation (hypoxia, TNF-$\alpha$, and $H_2O_2$). These stimuli converge in the production of reactive oxygen species (ROS), key signaling molecules during inflammation. Consistently, pre-treatment with the various stimuli strongly inhibited *Shigella* infection, already at early times post-infection (0.5 hpi; Figs 1E and F, and EV1H and I). The various compounds did not affect host cell viability, at the concentrations and incubation periods tested (Fig EV1J). It should be noted that host cells were extensively washed prior to infection to remove any remaining stressors. Moreover, arsenite or anisomycin treatment did not impair *Shigella* growth (Fig EV1K), thus excluding direct effects on the bacteria.

Importantly, treatment with the antioxidant N-acetyl-L-cysteine (NAC) reverted the inhibitory effect of arsenite or anisomycin on *Shigella* infection (Figs 1G and H, and EV1L), further confirming that the effect of these stressors on *Shigella* infection is mediated by oxidative stress. Overall, these results demonstrate that the response of epithelial cells to oxidative stress limits *Shigella* infection.

## *Shigella* binding to host cells is inhibited upon cellular stress

The evident inhibition of *Shigella* infection observed at 0.5 hpi strongly indicates that host cell stress affects the early steps of *Shigella* interaction with host cells. Accordingly, pre-treatment of cells with arsenite or anisomycin strongly inhibited *Shigella* binding to HeLa cells (*ca.* 10.0- and 2.0-fold compared to control, cfu, respectively; Fig 2A–C and Appendix Fig S1A–E). Comparable results were obtained in HCT-8 cells (Fig 2D and E, and Appendix Fig S1F). A possible effect of the stress inducers on actin cytoskeleton integrity and dynamics was excluded, since normal induction of actin-rich membrane ruffles by wild-type (WT) *Shigella* was observed in cells pre-treated with arsenite or anisomycin (white arrowheads in Fig 2A and Appendix Fig S1C, respectively). Accordingly, the efficiency of ruffle formation, i.e., the percentage of ruffles induced upon bacterial contact, was similar in cells treated with stressors and control cells (Appendix Fig S1G). To uncouple *Shigella* binding to host cells from subsequent steps of invasion, we performed parallel experiments with the *Shigella* ΔipaB mutant strain, which binds efficiently to host cells but is unable to invade (Menard *et al*, 1993). Treatment of cells with arsenite or anisomycin inhibited binding of *Shigella* ΔipaB, similar to the WT bacteria (Fig 2A–E, and Appendix Fig S1B, C, E and F). Binding of *Shigella* ΔicsA mutant strain to host cells was also inhibited by arsenite or anisomycin treatment (Appendix Fig S1H and I), demonstrating that the effect of the stressors is independent of the role of IcsA in *Shigella* adhesion (Brotcke Zumsteg *et al*, 2014). Overall, these results show that host cell stress has a strong inhibitory effect on *Shigella* binding.

Analysis of *Shigella* cell-to-cell spreading in control and arsenite or anisomycin-treated cells, by quantifying the area of *Shigella* infection foci using fluorescence microscopy and automated image analysis (Sunkavalli *et al*, 2017), excluded an effect of stress in the actin-based spreading of *Shigella* to neighboring cells (Appendix Fig S1J). In these experiments, infections were performed at different MOIs (MOI 10 for control, MOI 50 for anisomycin, and MOI 100 for arsenite), to achieve comparable levels of bacterial invasion.

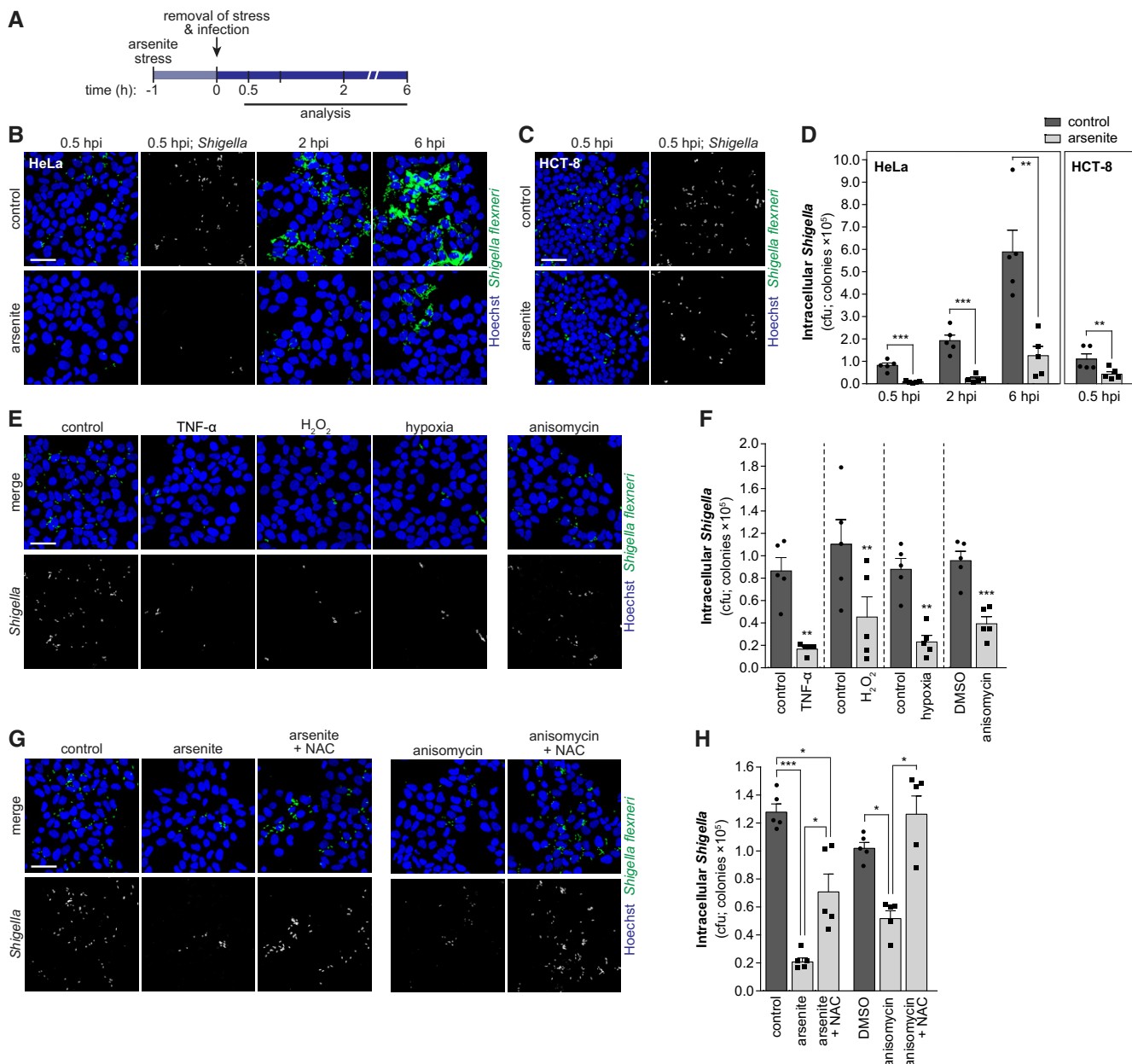

**Figure 1. *Shigella* infection is inhibited by host cell stress.**

A    Schematic representation of the experimental design.
B, C    Representative images of HeLa (B) or HCT-8 (C) cells infected with *Shigella* WT pre-treated or not with arsenite, analyzed at the indicated times post-infection.
D    Cfu quantification of intracellular bacteria in HeLa and HCT-8 cells pre-treated or not with arsenite and infected with *Shigella*.
E, F    Representative images (E) and cfu quantification (F) of intracellular bacteria in HeLa cells infected with *Shigella* WT after pre-treatment with TNF-α, H₂O₂, anisomycin, hypoxia, and corresponding controls, analyzed at 0.5 hpi.
G, H    Representative images (G) and cfu quantification (H) of intracellular *Shigella* in HeLa cells pre-treated with arsenite, anisomycin, stressors plus NAC, and corresponding controls.

Data information: *Shigella* infection was performed at MOI 10. Results are shown as mean ± s.e.m. of five independent experiments; *$P < 0.05$, **$P < 0.01$, ***$P < 0.001$ (*t*-test adjusted for multiple comparison for D—HeLa; paired *t*-test for D—HCT-8 and F; one-way ANOVA for H). Scale bars, 50 μm.

## ASM-dependent membrane remodeling upon stress inhibits *Shigella* binding

Considering that binding of *Shigella* to host cells is inhibited by stress and that this occurs in a relatively short timeframe (e.g.,

15 min pre-treatment with TNF-α; Figs 1E and F, and EV1H and I), we reasoned that the decreased binding of *Shigella* could result from modifications of the cellular membrane. In line with this possibility, membrane composition, specifically the presence of sphingolipids and cholesterol at the cell surface, has been shown to be required

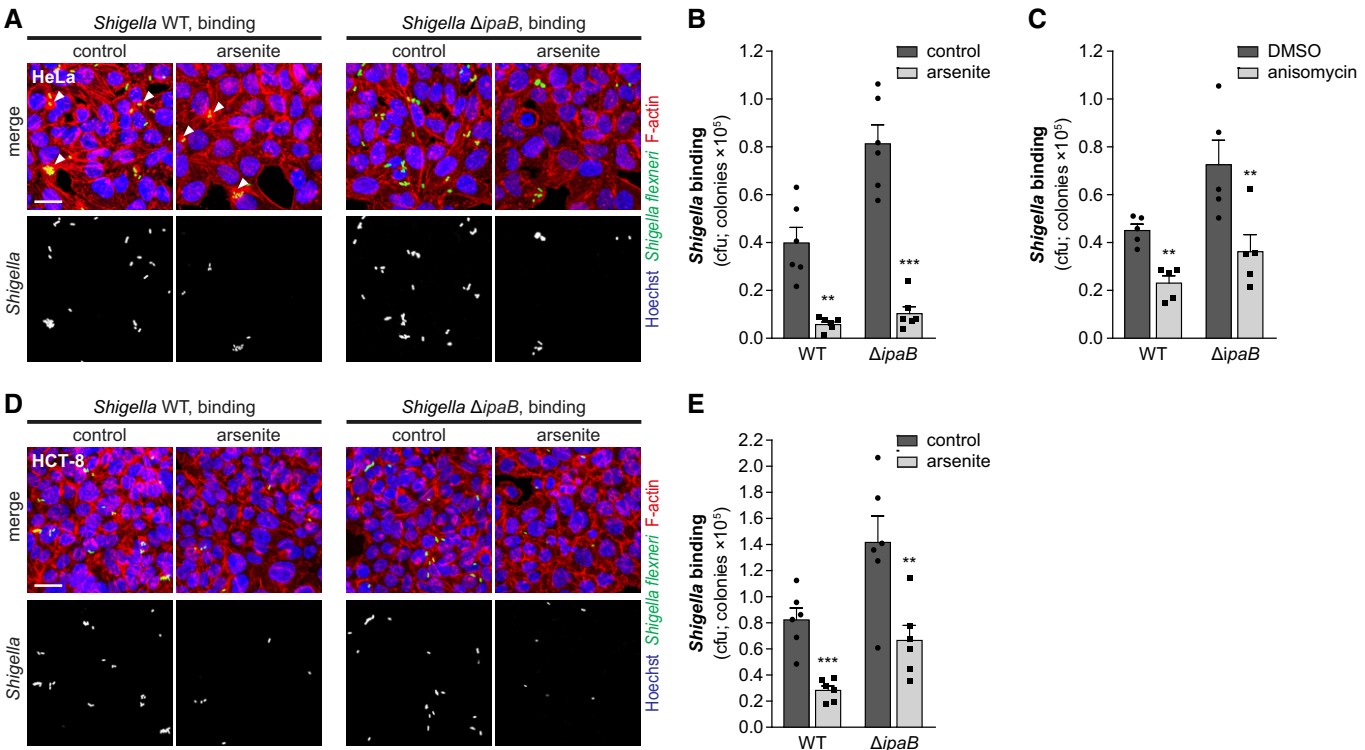

**Figure 2.  Host cellular stress inhibits *Shigella* binding to host cells.**

A, B   Representative images (A) and cfu quantification (B) of *Shigella* WT or Δ*ipaB* mutant strain bound to HeLa cells pre-treated or not with arsenite. Ruffle formation induced by *Shigella* WT in panel (A) is indicated by white arrowheads.

C   Cfu quantification of *Shigella* WT or Δ*ipaB* mutant strain bound to HeLa cells pre-treated with anisomycin or DMSO (control).

D, E   Representative images (D) and cfu quantification (E) of *Shigella* WT or Δ*ipaB* mutant bound to HCT-8 cells pre-treated or not with arsenite.

Data information: *Shigella* infection was performed at MOI 10 for *Shigella* WT or at MOI 50 for the Δ*ipaB* mutant strain; cells were incubated with the bacteria for 25 min. Results are shown as mean ± s.e.m. of 5 (C) or 6 (B, E) independent experiments; \*\**P* > 0.01, \*\*\**P* < 0.001 (paired *t*-test). Scale bars, 25 μm.

for successful *Shigella* infection (Lafont *et al*, 2002). The turnover of sphingomyelin, a ubiquitous sphingolipid component of animal cell membranes that is enriched in membrane rafts, is mediated by sphingomyelinases. These enzymes catalyze the breakdown of sphingomyelin to ceramide and phosphocholine (Goni & Alonso, 2002). Interestingly, neutral sphingomyelinase (NSM) and ASM are activated in response to various stress stimuli, including hydrogen peroxide, hypoxia, TNF-α, and infection (Hannun & Luberto, 2000; Grassme *et al*, 2003; Marchesini & Hannun, 2004).

To investigate whether the activation of sphingomyelinases and consequently the disruption of sphingolipid-rich membrane domains could account for the inhibition of *Shigella* binding to host cells observed upon stress, we used specific inhibitors of these enzymes. Inhibition of NSM by GW4869 did not rescue the impairment of *Shigella* infection prompted by arsenite or anisomycin (Fig EV2A–F). However, treatment with the ASM inhibitor amitriptyline partially reverted the inhibitory effect of arsenite on *Shigella* infection (Fig 3A–C) and fully reverted that caused by anisomycin (Fig 3D–F). Corroborating these results, knockdown of ASM blunted the effect of arsenite on *Shigella* infection (Fig 3G–I). Of note, in the absence of stress, the inhibition of ASM activity by amitriptyline or ASM knockdown did not affect *Shigella* infection (Fig EV2G–J). The effect of amitriptyline in blunting ASM enzymatic activity and ceramide production upon arsenite or anisomycin treatment was confirmed (Fig EV2K–M). In

addition, the decreased ASM activity, RNA, and protein levels upon knockdown were confirmed (Fig EV2N–P). Taken together, these results show that, in conditions of cellular stress, ASM activity is responsible for the inhibition of *Shigella* infection.

Acid sphingomyelinase is usually associated with the lysosomal compartment, but, upon activation, it is redistributed to the outer leaflet of the plasma membrane, where it hydrolyzes sphingomyelin giving rise to ceramide-rich platforms (Grassme *et al*, 2001). To validate whether ASM is relocalized as part of the cell response to stress, we used confocal microscopy followed by 3D reconstruction to visualize the accumulation of ASM and ceramide at the membrane. Indeed, treatment of cells with arsenite or anisomycin induced a marked accumulation of ASM at the cell surface, in both HeLa and HCT-8 cells (Fig 3J and K, Appendix Fig S2A—HeLa; Appendix Fig S2E—HCT-8). Consistent with these findings, a strong accumulation of ceramide was also observed at the membrane of cells treated with arsenite or anisomycin (Fig 3L and M, Appendix Fig S2C—HeLa; Appendix Fig S2F—HCT-8). As expected, ASM knockdown diminished ceramide accumulation in response to arsenite (Appendix Fig S2D). The ASM staining specificity was confirmed in cells transfected with ASM siRNA, in which a considerable decrease in ASM foci was observed (Appendix Fig S2B). To further reinforce these observations, we quantified ceramide in live cells, by flow cytometry. In live cells, the ceramide antibody can only recognize extracellular/

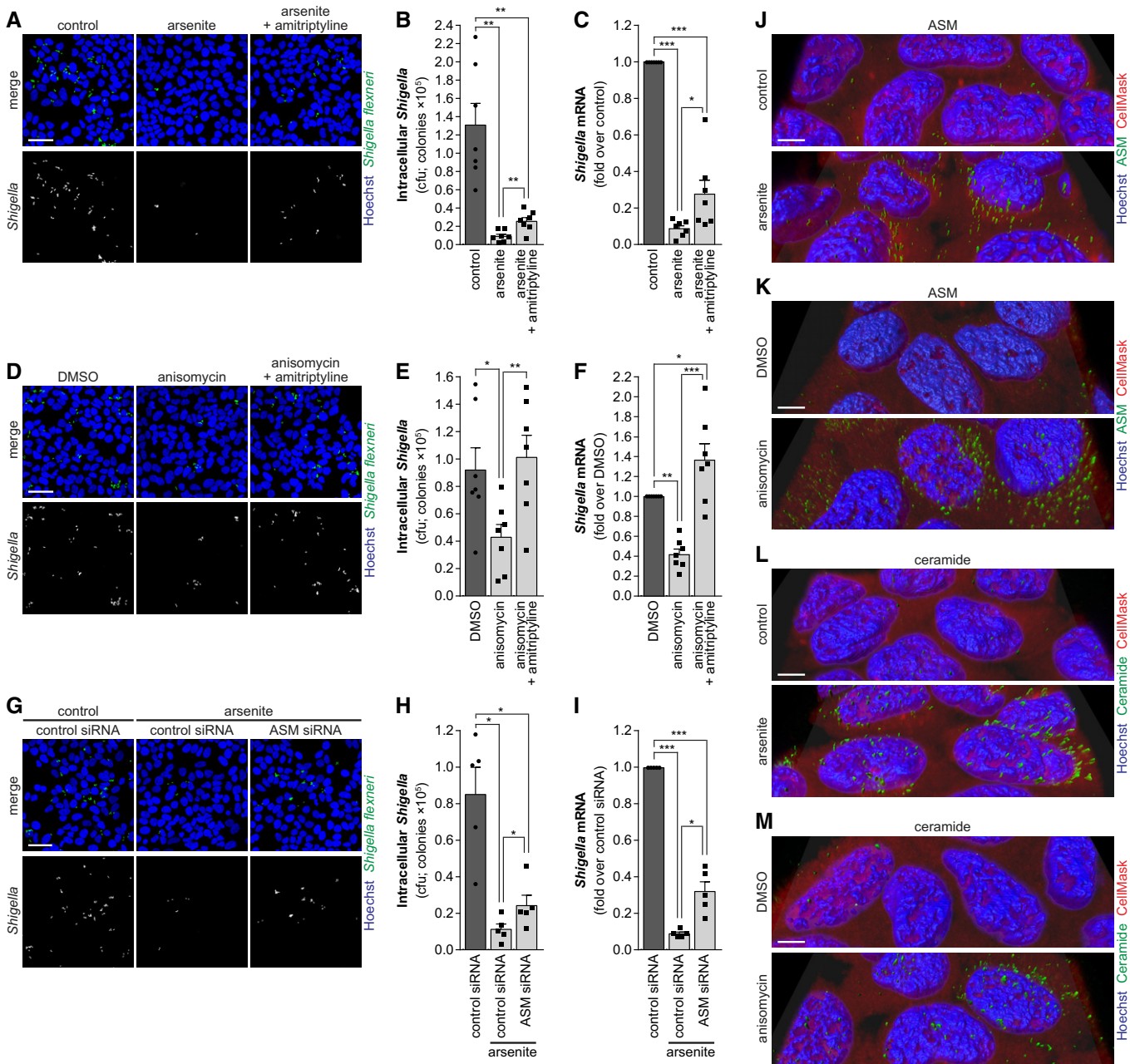

**Figure 3.  Inhibition of *Shigella* infection upon host cell stress is a consequence of ASM activation.**

A–F    Representative images (A, D), cfu (B, E), and qRT–PCR (C, F) quantification of intracellular *Shigella* in HeLa cells pre-treated with arsenite or anisomycin, in the presence or not of amitriptyline, and corresponding controls.

G–I    Representative images (G), cfu (H), and qRT–PCR (I) quantification of intracellular *Shigella* in HeLa cells transfected with ASM siRNA or control siRNA and pre-treated or not with arsenite prior to infection.

J–M    3D reconstruction of representative images of ASM (J, K) or ceramide (L, M) staining in HeLa cells treated with arsenite or anisomycin and corresponding controls. ASM, ceramide, and Hoechst staining were surface-converted by voxel distance.

Data information: Infection was performed with *Shigella* WT at MOI 10 and analyzed at 0.5 hpi. Results are shown as mean ± s.e.m. of 5 (H, I) or 7 (B–F) independent experiments; *P < 0.05, **P < 0.01, ***P < 0.001 (one-way ANOVA). Scale bars, 50 μm (A, D, G) and 5 μm (J–M).

exposed epitopes and thus exclusively labels ceramide present at the cellular surface. This analysis confirmed a significant increase of ceramide in cells treated with arsenite and anisomycin, and a reversion of the increase upon amitriptyline treatment (Fig EV2L and M).

Overall, these results show that the binding of *Shigella* to host cells in conditions of cellular stress is impaired as a consequence of the membrane remodeling induced by ASM activation and translocation to the plasma membrane.

## p38 MAPK is required for the inhibition of *Shigella* infection upon stress

Previous studies have shown a strong relation between sphingomyelinases and p38 mitogen-activated protein kinase (MAPK) activity. Oxidative stress leads to the activation of sphingomyelinases and ceramide production, which activates p38 MAPK pathway (Chen *et al*, 2008). In turn, p38 MAPK was shown to activate ASM in glial cells (Bianco *et al*, 2009). Thus, we hypothesized that the p38 MAPK pathway could be an important player in the activation of the sphingomyelinases and consequent inhibition of *Shigella* infection, in a context of cellular stress. Firstly, we confirmed that the various stressors (arsenite, anisomycin, TNF-α, and $H_2O_2$) activate the p38 MAPK pathway, by Western blotting for the active phosphorylated form of p38 (Fig 4A). Supporting our initial hypothesis, inhibition of p38 MAPK pathway by siRNA

knockdown (Mapk14) or treatment with SB203580 reverted the inhibitory effect of arsenite (Figs 4B–D and EV3A) or anisomycin (Figs 4E and F, and EV3B and C) on *Shigella* infection. Importantly, we verified that inhibition of p38 MAPK pathway, by treatment with SB203580, inhibited the production of ceramide induced by arsenite and anisomycin (Figs 4G and H, and EV3D and E), demonstrating that p38 MAPK pathway is required for ASM activation and membrane remodeling upon stress. The effective inhibition of the p38 MAPK pathway by SB203580 upon arsenite or anisomycin treatment, and the knockdown of p38 were confirmed by Western blotting (Fig EV3F–H). p38 knockdown or SB203580 treatment had no effect on *Shigella* infection in the absence of stress (Fig EV3I and J).

Taken together, these results demonstrate that the inhibition of *Shigella* infection during stress relies on the activation of the p38 MAPK pathway.

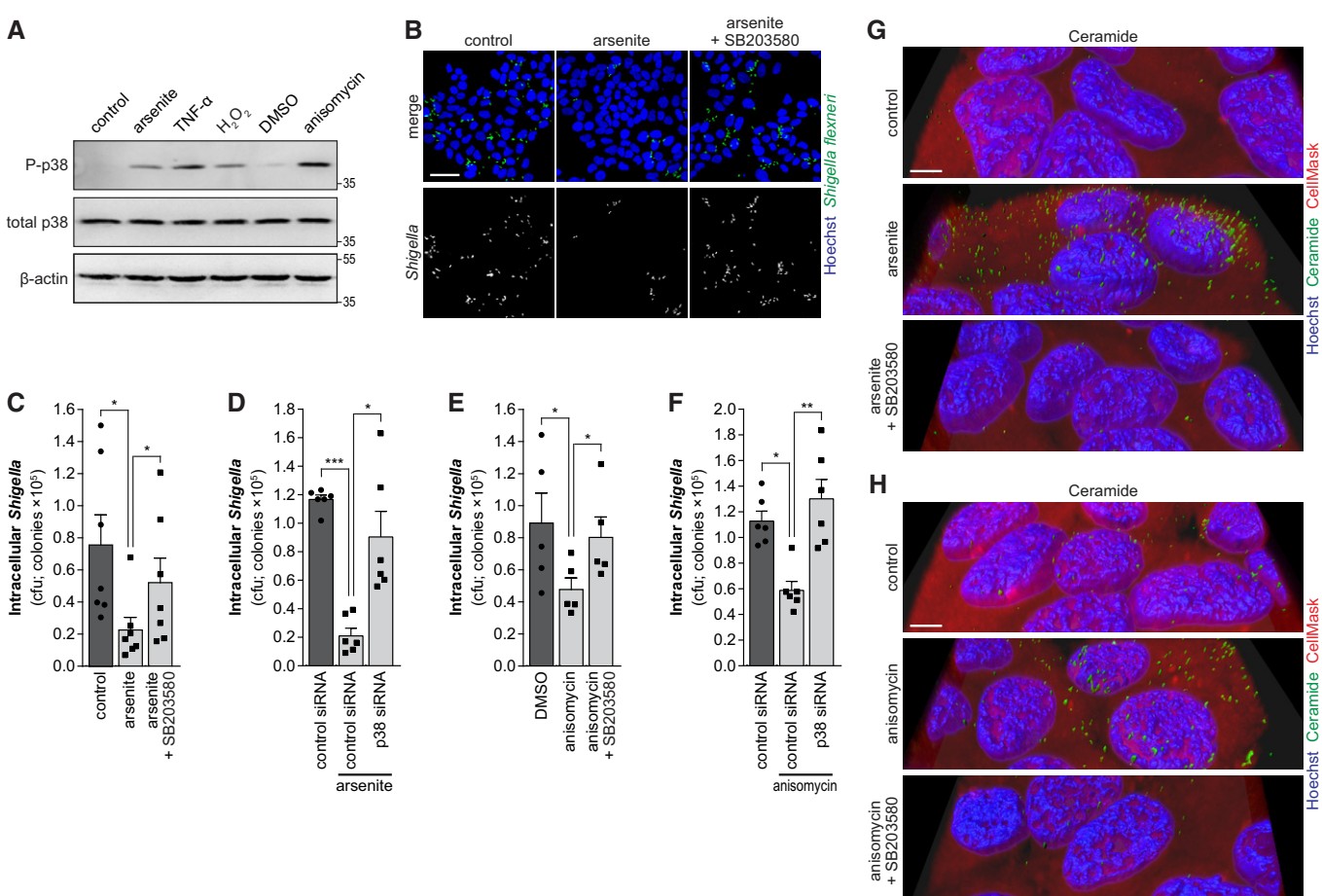

**Figure 4.   MAPK p38 inhibition restores *Shigella* infection of host cells exposed to stress.**

A       Western blot analysis of p38 MAPK phosphorylation in HeLa cells treated with arsenite, TNF-α, $H_2O_2$, or anisomycin; β-actin was used as loading control.

B–F    (B, C, and E) Representative images (B) and cfu quantification (C, E) of intracellular *Shigella* in HeLa cells pre-treated with arsenite or anisomycin, in the absence or presence of the p38 inhibitor SB203580, and corresponding controls. (D and F) Cfu quantification of intracellular *Shigella* in HeLa cells transfected with p38 siRNA or control siRNA and pre-treated or not with arsenite (D) or anisomycin (F) prior to infection.

G, H   3D reconstruction of representative images of ceramide staining in HeLa cells treated with arsenite (G) or anisomycin (H), in the presence or absence of SB203580, and corresponding controls. Ceramide and Hoechst staining were surface-converted by voxel distance.

Data information: Infection was performed with *Shigella* WT at MOI 10 and analyzed at 0.5 hpi. Results are shown as mean ± s.e.m. of 5 (E), 6 (D and F), or 7 (C) independent experiments; *$P < 0.05$, **$P < 0.01$, ***$P < 0.001$ (one-way ANOVA). Scale bars, 50 μm (B) and 5 μm (G, H).
Source data are available online for this figure.

**Figure 5.   *Salmonella* motility compensates for membrane remodeling in cells exposed to stress.**

A–C   Representative images (A), cfu (B), and qRT–PCR (C) quantification of intracellular *Salmonella* in HeLa cells pre-treated with arsenite or anisomycin and corresponding controls. Infection was performed with *Salmonella* WT and analyzed at 0.5 hpi.

D     Representative images of bacteria bound to HeLa cells pre-treated or not with arsenite followed by incubation with *Salmonella* WT or Δ4 mutant strain for 10 min.

E, F   Violin plots showing the distribution of the number of bacteria bound per infected cell in HeLa cells pre-treated or not with arsenite followed by incubation with *Salmonella* WT (E) or Δ4 mutant strain (F) for 10 min. Results are shown for 50 infected cells per condition and independent experiment (total 250 cells); white circles show the medians, box limits indicate the 25[th] and 75[th] percentiles, whiskers extend 1.5 times the interquartile range from the 25[th] and 75[th] percentiles; polygons extend to extreme values.

G, H   Representative images (G) and cfu quantification (H) of bacteria bound to HeLa cells pre-treated or not with arsenite followed by incubation with *Salmonella* mutant strains Δ*fliC*, Δ*fliC*/pFliC, or Δ*flhC* for 10 min.

I, J   Representative images (I) and cfu quantification (J) of *Listeria* bound to HeLa cells pre-treated or not with arsenite followed by incubation with the bacteria for 20 min.

K, L   Representative images (K) and cfu quantification (L) of *Yersinia* bound/internalized by HeLa cells pre-treated or not with arsenite followed by 70-min incubation with the bacteria.

Data information: Infection was performed at MOI 25 (A–C) or MOI 50 (D–H) for *Salmonella*, and MOI 50 for *Listeria* (I, J) and *Yersinia* (K, L). Results are shown as mean ± s.e.m. of 5 (C, E, F, H), 6 (B—arsenite and L), or 7 (B—anisomycin and J) independent experiments; ***$P$ < 0.001 (paired $t$-test for B, C, J, L; Mann–Whitney $U$-test for E, F; two-way ANOVA for H). Scale bars, 50 μm (A) and 25 μm (D, G, I, K).

## Bacterial motility overcomes the restriction posed by stress-induced host cell membrane remodeling

Membrane composition has been shown to play critical roles for the adhesion and invasion of various bacterial pathogens in addition to *Shigella*, including the closely related pathogen *Salmonella* Typhimurium (Garner *et al*, 2002; Santos *et al*, 2013). We thus reasoned that the stress-induced membrane remodeling could likewise affect *Salmonella* infection. Surprisingly, however, *Salmonella* infection at 0.5 hpi was normal in cells pre-treated with arsenite or anisomycin (Fig 5A–C), suggesting that *Salmonella* is able to cope with the remodeled host cell membrane. Interestingly, the pattern of infection was altered in cells treated with the stress inducers, showing fewer infected cells, but more bacteria per infected cell (Fig 5A). This observation was further strengthened by the results of *Salmonella* binding assays, in which a significant increase in the number of bacteria per infected cell was observed in cells pre-exposed to arsenite or anisomycin, compared to control cells (Fig 5D and E, and Appendix Fig S3A). To confirm that this phenotype was related to bacterial binding, we used an invasion-deficient but binding proficient *Salmonella* mutant strain (*Salmonella* Δ4), which lacks four effector proteins (SopE, SopE2, SopA, and SipB) essential for invasion (Schlumberger *et al*, 2005; Lara-Tejero & Galan, 2009; Misselwitz *et al*, 2011b). Similar to WT *Salmonella*, in conditions of cellular stress the Δ4 mutant strain infected a lower number of cells than in the absence of stress, but also accumulated in specific sites, resulting in a higher number of bacteria per infected cell (Fig 5D and F, and Appendix Fig S3A). Moreover, a direct effect of stress on *Salmonella* intracellular replication was excluded, given that arsenite treatment post-invasion did not affect infection (Appendix Fig S3B and C). These results indicate that, although the total bacterial load is similar, in cells subjected to stress there are fewer *Salmonella*-infected cells, though exhibiting a higher number of bacteria bound per cell.

The patchy pattern of *Salmonella* binding to cells exposed to stress suggests that the bacteria are able to accumulate in the remaining permissive binding sites after stress-induced membrane remodeling has occurred. In contrast to *Shigella*, *Salmonella* is a motile pathogen, and therefore, a possible explanation for this phenomenon is that *Salmonella* can compensate for the lack of permissive sites by scanning the cellular surface through a process of near cellular surface swimming, enabled by its flagellar motility (Misselwitz *et al*, 2012). Most *Salmonella* serovars, including Typhimurium, have two flagellin genes, *fliC* and *fljB*, allowing antigen switches between two alternative forms of its flagellin filament protein (Andrewes, 1922). To test the relevance of bacterial motility in conditions of host cell stress, we used the *Salmonella* motility-deficient mutant strain Δ*fliC*, which lacks the *fliC* gene but retains expression of FljB. However, the frequency of flagellin switching is low [approximately 1/5,000 per cell per generation, biased toward the *fljB*[OFF] orientation (Gillen & Hughes, 1991)], and thus, the *Salmonella* Δ*fliC* mutant strain is essentially non-motile. Additionally, we used a completely non-motile *Salmonella* mutant (Δ*flhC*), due to an insertion of the MudJ transposon into the flagella master regulator *flhC* gene. Remarkably, arsenite and anisomycin pre-treatment inhibited very efficiently the binding of the non-motile *Salmonella* Δ*fliC* and Δ*flhC* mutants (Fig 5G and H, and Appendix Fig S3D and E). Complementation of the Δ*fliC* mutant, by ectopic expression of FliC from a native promoter, restored the ability of *Salmonella* to bind to cells subjected to stress, at levels comparable to that of control cells (Fig 5G and H, and Appendix Fig S3D and E), demonstrating causality between bacterial motility and the ability to overcome membrane remodeling. Based on these observations, we next addressed the effect of host stress on the infection by two additional pathogens, *L. monocytogenes* and *Yersinia pseudotuberculosis*, which repress flagella at 37°C and are thus non-motile when grown in these conditions (Peel *et al*, 1988; Kapatral & Minnich, 1995). In agreement with the results reported above for *Shigella* and *Salmonella*, arsenite treatment inhibited the infection by *Listeria* and *Yersinia* (Fig 5I–L), demonstrating that membrane remodeling generally impairs infection by non-motile bacterial pathogens.

Overall, these results show that motility is a strong determinant of binding of bacteria to cells exposed to stress. Under these conditions, while binding of non-motile bacteria to host cells is strongly compromised, motile bacteria are able to efficiently sample the host cellular surface and accumulate at the remaining permissive sites for efficient bacterial invasion.

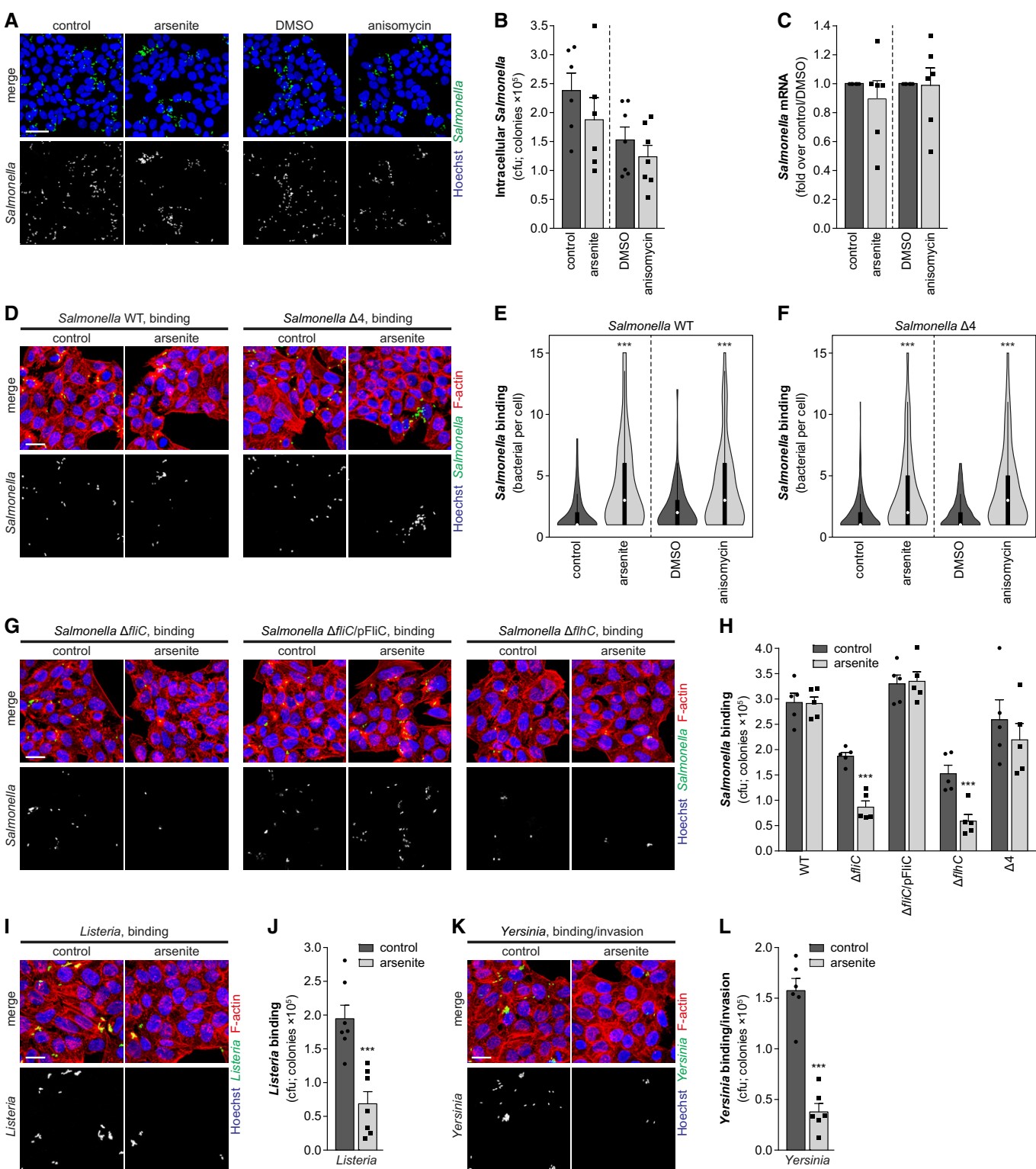

**Figure 5.**

## *Shigella* intracellular replication inhibits re-infection by non-motile bacteria

Given that *Shigella* intracellular replication causes stress, entailing oxidative stress, cytokine secretion, and MAPK activation (Pedron

*et al*, 2003; Carneiro *et al*, 2009; Kasper *et al*, 2010), and that the host cell response to stress ultimately inhibits *Shigella* binding, we hypothesized that cells infected with *Shigella* could become refractory to re-infection. To test this possibility, we designed a re-infection assay in which mock-treated or cells infected with

**Figure 6.   Re-infection by non-motile bacteria is inhibited by *Shigella* intracellular replication.**

A    Schematic representation of the experimental design for re-infection experiments. Cells were infected with *Shigella*-mCherry (primary infection) or mock-treated. At 3 hpi, cells were re-infected with *Shigella* or *Salmonella* expressing GFP (secondary infection). Secondary infection was analyzed at 0.5 hpi (i.e., 1 h after addition of bacteria to cells).

B–E    Representative images (B, D) and cfu quantification (C, E) of re-infection assays with *Shigella* WT (B, C) or *Salmonella* WT (D, E) of HeLa cells primarily infected with *Shigella* WT or Δ*ipaB*/Inv mutant strain, or mock-treated. Secondary infection is shown in bottom images.

F, G    Representative images (F) and cfu quantification (G) of the re-infection assays with *Salmonella* Δ*fliC*, Δ*fliC*/pFliC, or Δ*flhC* mutant strains in HeLa cells primarily infected with *Shigella* WT or mock-treated. Secondary infection is shown in bottom images.

H    Schematic representation of the experimental design for re-infections upon knockdown or inhibitor treatment.

I, J    Representative images (I) and cfu quantification (J) of the secondary infection with *Shigella* WT, in cells transfected with ASM siRNA or control siRNA.

K–M    Cfu quantification of the secondary infection with *Shigella* WT in HeLa cells following amitriptyline (K), transfection with MAPK p38 siRNA (L), or SB203580 treatment (M).

Data information: Infection was performed at MOI 10 for *Shigella* WT, MOI 350 for *Shigella* Δ*ipaB*/Inv, MOI 25 for *Salmonella* WT (D, E, G), or MOI 50 for *Salmonella* WT, Δ*fliC*, Δ*fliC*/pFliC, or Δ*flhC* mutant strains (F, G). Results are shown as mean ± s.e.m. of five independent experiments; *$P < 0.05$, **$P < 0.01$, ***$P < 0.001$ (one-way ANOVA for C, E, J-M; two-way ANOVA for G). Scale bars, 50 μm.

*Shigella*-mCherry for 3 h (primary infection) were re-infected with *Shigella*-GFP (secondary infection; Fig 6A). Interestingly, the secondary infection with *Shigella*-GFP was significantly decreased in cells previously infected with *Shigella*-mCherry, when compared to mock-treated cells (Figs 6B and C, and EV4A). To test whether the internalization of a replication-deficient *Shigella* mutant strain is sufficient to block re-infection, we performed parallel experiments using the *Shigella*-mCherry Δ*ipaB*/Inv strain for the primary infection. The *Shigella* Δ*ipaB*/Inv strain has a compromised T3SS secretion, though it is internalized due to the expression of the *Yersinia* invasin protein (Isberg *et al*, 1987). Once inside cells, this mutant is unable to escape the internalization vacuole and does not replicate (Suzuki *et al*, 2006). Interestingly, we observed that primary infection with this mutant strain did not compromise re-infection, which was comparable to that observed in mock-treated cells (Figs 6B and C, and EV4A). Taken together, these observations demonstrate that *Shigella* intracellular replication and/or presence of the bacteria in the host cytoplasm is required for inhibition of re-infection.

In agreement with the results described above showing that host response to stress does not restrict infection by motile bacteria, infection with *Salmonella* WT-GFP (secondary infection) was not inhibited in cells previously infected with *Shigella*-mCherry (Figs 6D and E, and EV4B). Similar results were obtained in HCT-8 cells; specifically, we observed that primary infection with *Shigella*-mCherry inhibits re-infection by *Shigella*-GFP (Fig EV4E–G), but not by *Salmonella*-GFP (Fig EV4H–J). Of note, the primary infection with *Shigella*-mCherry WT or Δ*ipaB*/Inv mutant strain was comparable in the different experimental conditions, both in HeLa and in HCT-8 cells (Fig EV4C, D, K and L). In agreement with the lack of membrane remodeling observed upon *Salmonella* infection (Appendix Fig S3F), the primary infection with *Salmonella*-mCherry did not affect re-infection by *Shigella*-GFP (Fig EV4M and N).

To evaluate whether the motility of *Salmonella* is related to its ability to circumvent the restriction to infection caused by *Shigella* intracellular replication, we performed re-infection assays using the *Salmonella* motility mutants described above. Infection with *Salmonella* Δ*fliC* and Δ*flhC* was inhibited in cells containing replicating *Shigella*-mCherry (Fig 6F and G), whereas infection with the Δ*fliC*/pFliC complemented strain was not affected (Fig 6F and G). This clearly reinforces the notion that the motility of *Salmonella* is the decisive feature to overcome the inhibition posed by the cellular response to *Shigella* infection. Of note, we confirmed that the levels

of the primary *Shigella*-mCherry infection were similar in the experiments with the different *Salmonella* strains, excluding any indirect effect on the primary infection (Fig EV4O).

Collectively, these results demonstrate that *Shigella* intracellular replication in epithelial cells inhibits subsequent binding and re-infection by *Shigella* and, presumably, of other non-motile bacteria.

### *Shigella* intracellular replication induces host cell membrane remodeling *in vitro* and *in vivo*

Next, we pondered whether *Shigella* infection could cause membrane remodeling, and thus explain the inhibition of re-infection. To address this question, we either performed the re-infection experiment in ASM knockdown cells or used amitriptyline to inhibit ASM activity in cells at 2 hpi with *Shigella*, proceeding with re-infection with *Shigella*-GFP after 1 h of amitriptyline treatment (Fig 6H). Under these conditions, secondary infection by *Shigella*-GFP was not inhibited by the primary infection (Figs 6I–K and EV5A–C). Similarly, inhibition of the p38 MAPK pathway achieved by p38 knockdown or treatment with SB203580 (Figs 6L and M, and EV5D and E), as well as oxidative stress mitigation by NAC treatment (Fig EV5F and G), dampened the inhibitory effect of the primary infection. Of note, treatment with amitriptyline, SB203580, NAC, as well as ASM or p38 knockdown, had no effect on the primary infection by *Shigella*-mCherry (Fig EV5H–N). Overall, these results imply that ASM-mediated membrane remodeling, induced by oxidative stress-activated p38 MAPK pathway, is responsible for the inhibition observed in the re-infection experiments reported above.

To provide direct evidence that membrane remodeling occurs as a consequence of the host cell response to *Shigella* infection, we measured ASM activity and analyzed the accumulation of ASM and ceramide at the membrane of *Shigella*-infected cells. In agreement with an involvement of ASM in this process, we observed a clear accumulation of ASM in the membrane of *Shigella* WT-infected cells (Figs 7A and EV5O), alongside a clear increase of ceramide (Figs 7B and EV5P); this phenotype was not observed in *Shigella* Δ*ipaB*/Inv (Fig EV5R and S), which is in perfect agreement with the observation that primary infection with this mutant strain does not compromise re-infection. Interestingly, both ASM and ceramide accumulated to a higher extent in bystander cells, compared to cells with intracellular bacteria (Fig 7A and B, and EV5O–Q). Enzymatic assays on membrane fractions from *Shigella*-infected cells (3 hpi)

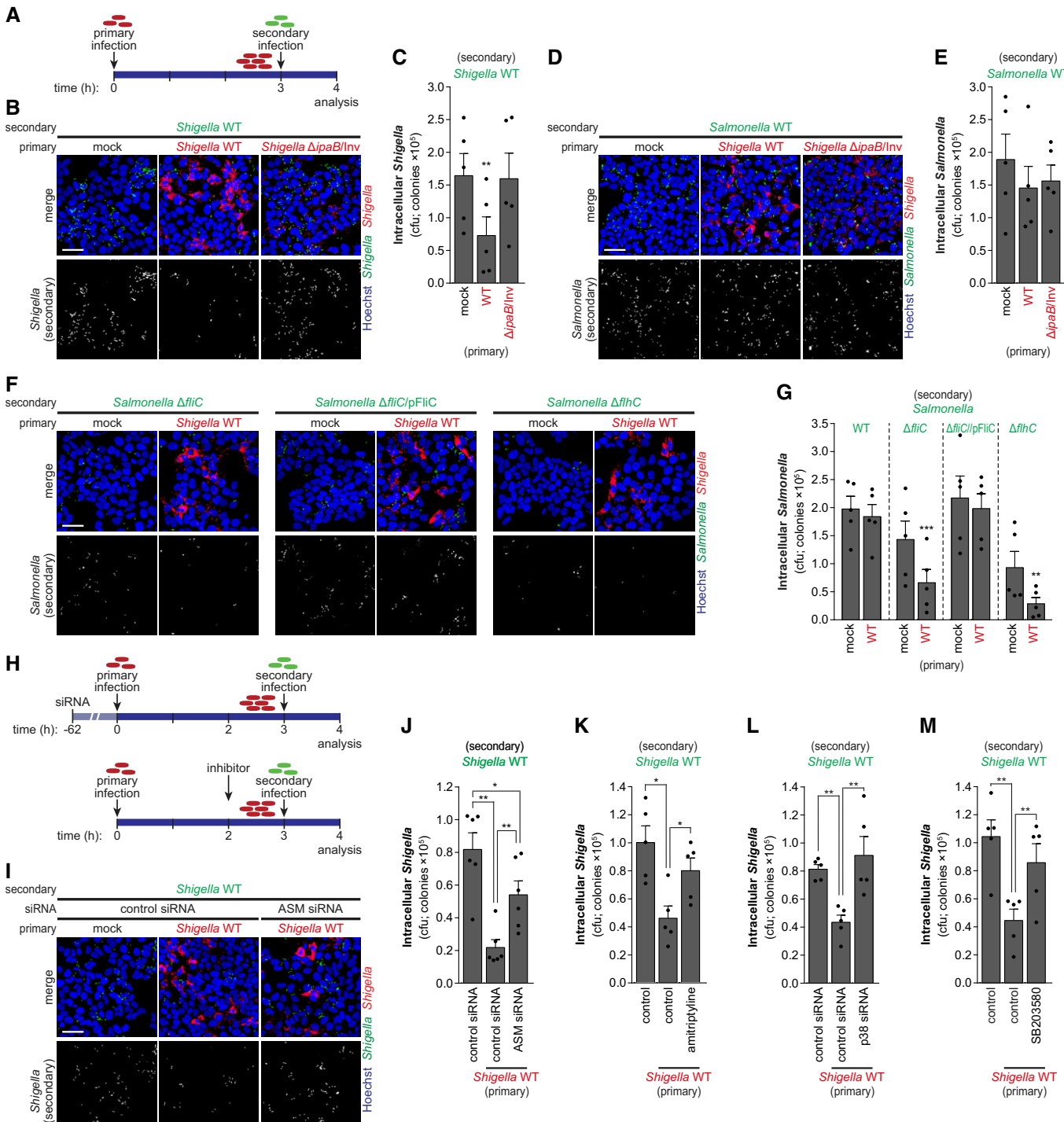

**Figure 6.**

showed a modest but significant increase in the activity of ASM (Fig 7C), correlating with a clear increase in ceramide levels (Fig 7D).

Collectively, our results demonstrate that *Shigella* intracellular replication in epithelial cells leads to the activation of ASM, resulting in formation of ceramide domains, which impairs re-infection by *Shigella*.

Having shown that *Shigella* infection induces major changes in membrane composition *in vitro*, we wondered whether this process could be observed *in vivo*. For this purpose, we used a guinea pig model, a well-established animal model for the study of *Shigella* infection (Arena *et al*, 2015), and compared the ceramide accumulation in colon tissue samples from non-infected and *Shigella*-infected animals. In complete agreement with the *in vitro* data, we observed

**Figure 7.  Intracellular *Shigella* replication induces remodeling of the host cell plasma membrane *in vitro* and *in vivo*.**

A, B   3D reconstruction of representative images of ASM (A) or ceramide (B) staining in HeLa cells infected with *Shigella* WT or mock-treated, analyzed at 3 hpi. ASM, ceramide, Hoechst, and *Shigella* staining were surface-converted by voxel distance.

C, D   ASM enzymatic activity (C) and ceramide (D) measurement in HeLa cells infected with *Shigella* WT or mock-treated, analyzed at 3 hpi. The ASM enzymatic activity was determined in the membrane fraction corresponding to $3.0 \times 10^5$ cells per condition. Ceramide is shown normalized to mock-treated cells.

E–H   Representative spinning-disk confocal images of colon tissue sections of non-infected (E and F) and *Shigella*-infected (G and H) guinea pigs, collected at 4 hpi. Panels (E and G) are montages of multiple image fields; dashed boxes in (E and G) are shown enlarged in (F and H), respectively.

I   Mean fluorescence intensity of ceramide signal in surface epithelial cells (outermost tissue layer) of non-infected or *Shigella*-infected guinea pig colon tissue sections.

J   Model depicting the effect of stress-induced host plasma membrane remodeling on infection by bacterial pathogens. Different stressors, including TNF-α stimulation and *Shigella* intracellular replication, induce the relocalization and activation of acid sphingomyelinase (ASM) to the host plasma membrane, leading to the conversion of sphingomyelin into ceramide. As a consequence of membrane remodeling, the binding of non-motile bacteria, such as *Shigella*, to host cells is inhibited. Motile bacteria, such as *Salmonella*, are able to scan the host membrane for the remaining permissive binding sites, and thus circumvent this defense mechanism.

Data information: *Shigella* infection was performed at MOI 100 (A, B). Results are shown as mean ± s.e.m. of 9 (C) or 6 (D) independent experiments; whiskers in the box-plot (I) correspond to min/max from 10 (non-infected) or 13 (*Shigella*) tissue sections. *$P < 0.05$, **$P < 0.01$, ***$P < 0.001$ (one-way ANOVA for C, D; unpaired *t*-test for I). Scale bars, 5 μm (A, B), 50 μm (E, G), and 12.5 μm (F, H).

a clear accumulation of ceramide in colon tissue samples from *Shigella*-infected animals (4 hpi of challenge with *Shigella* WT) compared to non-infected controls (Fig 7E–I). Interestingly, the enhanced ceramide signal in infected samples was clearly more noticeable in bystander cells than in cells with intracellular bacteria.

Overall, these results strongly corroborate the notion that *Shigella* intracellular replication, both *in vitro* and in an animal model of *Shigella* infection, leads to the activation of ASM and consequent membrane remodeling.

# Discussion

Although intestinal epithelial cells are now considered crucial players in the defense against infection by bacterial pathogens, exactly how their general stress response helps to fend off potential bacterial invaders is incompletely understood. In this study, we reveal that epithelial cell stress generated by inflammatory cues and oxidative insults, namely arsenite, anisomycin, hypoxia, hydrogen peroxide, and cytokines, causes a strong inhibition of *Shigella* binding to host cells. Mechanistically, we determined that this inhibition results from an extensive remodeling of the host plasma membrane due to stress-induced activation of ASM that is enforced by p38 MAPK activation. This depletes permissive binding sites on the plasma membrane drastically and so hinders *Shigella*'s attempts to bind and, ultimately, invade host cells (Fig 7J).

The relevance of ASM activity and ensuing membrane remodeling in the context of infection by diverse bacterial pathogens has been studied before. ASM is required for efficient infection by *Neisseria gonorrhoeae* and *Neisseria meningitidis* (Grassme *et al*, 1997; Simonis *et al*, 2014; Faulstich *et al*, 2015), whereas for other pathogens, ASM is involved in the host defense against infection. For example, ASM KO mice are much more susceptible to *L. monocytogenes* infection than wild-type animals (Utermohlen *et al*, 2003), a phenotype that was linked with lysosomal dysfunction in macrophages (Utermohlen *et al*, 2003, 2008; Schramm *et al*, 2008). Based on the results described herein, we suggest that the higher susceptibility of ASM KO mice to *Listeria* infection can also be linked with the role of ASM in the remodeling of epithelial cell membrane to restrict adhesion and entry of non-motile pathogens. It is conceivable that the absence of ASM precludes membrane remodeling

events that would normally occur, effectively increasing the number of permissive sites on the cell surface. In agreement with this hypothesis, the entry of *Listeria* into host cells was shown to require the localization of the host receptors E-cadherin and HGF-R/Met in specific membrane domains (Seveau *et al*, 2004). Of note, *Listeria* should be considered a non-motile pathogen in the human host due to the temperature-dependent expression of flagella, which is strongly reduced at mammalian physiological temperature [37°C (Peel *et al*, 1988)] and therefore likely to be strongly affected by host plasma membrane remodeling (see below).

Noticeably, ASM KO mice are also more susceptible to *Salmonella* infection than wild-type animals, albeit less so than to *Listeria* (Utermohlen *et al*, 2003). The apparent discrepancy with our results (showing that *Salmonella* binding to epithelial cells is not affected by ASM activation) can likely be explained by the fact that the reported increased susceptibility of ASM KO mice to *Salmonella* derives from an impaired ability of ASM KO macrophages to kill *Salmonella* (McCollister *et al*, 2007). Nonetheless, we cannot exclude the possibility that *in vivo* the inflammatory conditions associated with *Salmonella* infection can induce a stronger effect at the level of membrane remodeling leading to inhibition of *Salmonella* invasion of epithelial cells.

Our results reinforce the relevance of the flagellar motility as a crucial mechanism used by bacteria to reach preferred sites of infection, particularly when these are scarcely available, such as under stress conditions. This is well illustrated by the ability of *Salmonella*—a motile pathogen—to overcome host defenses based on host membrane remodeling, which are thus particularly detrimental to non-motile pathogens. Here, we show that global binding and invasion of *Salmonella* are unaffected by stress, since flagellar motility enables this bacterium to scan the host cellular surface and accumulate at the remaining permissive entry sites. Conversely, non-motile *Salmonella* mutants show impaired binding to host cells in stress conditions. These *in vitro* observations are in agreement with previous reports showing that mutations affecting *Salmonella* motility impair bacterial colonization of the inflamed mouse intestine, but not of the normal gut (Stecher *et al*, 2004, 2008). This phenotype is, at least in part, attributed to flagella-facilitated chemotactic movement, which facilitates access to the nutrient-rich molecules released as part of the mucosal defense during inflammation (Stecher *et al*, 2008). However, our findings showing that host membrane remodeling in response to

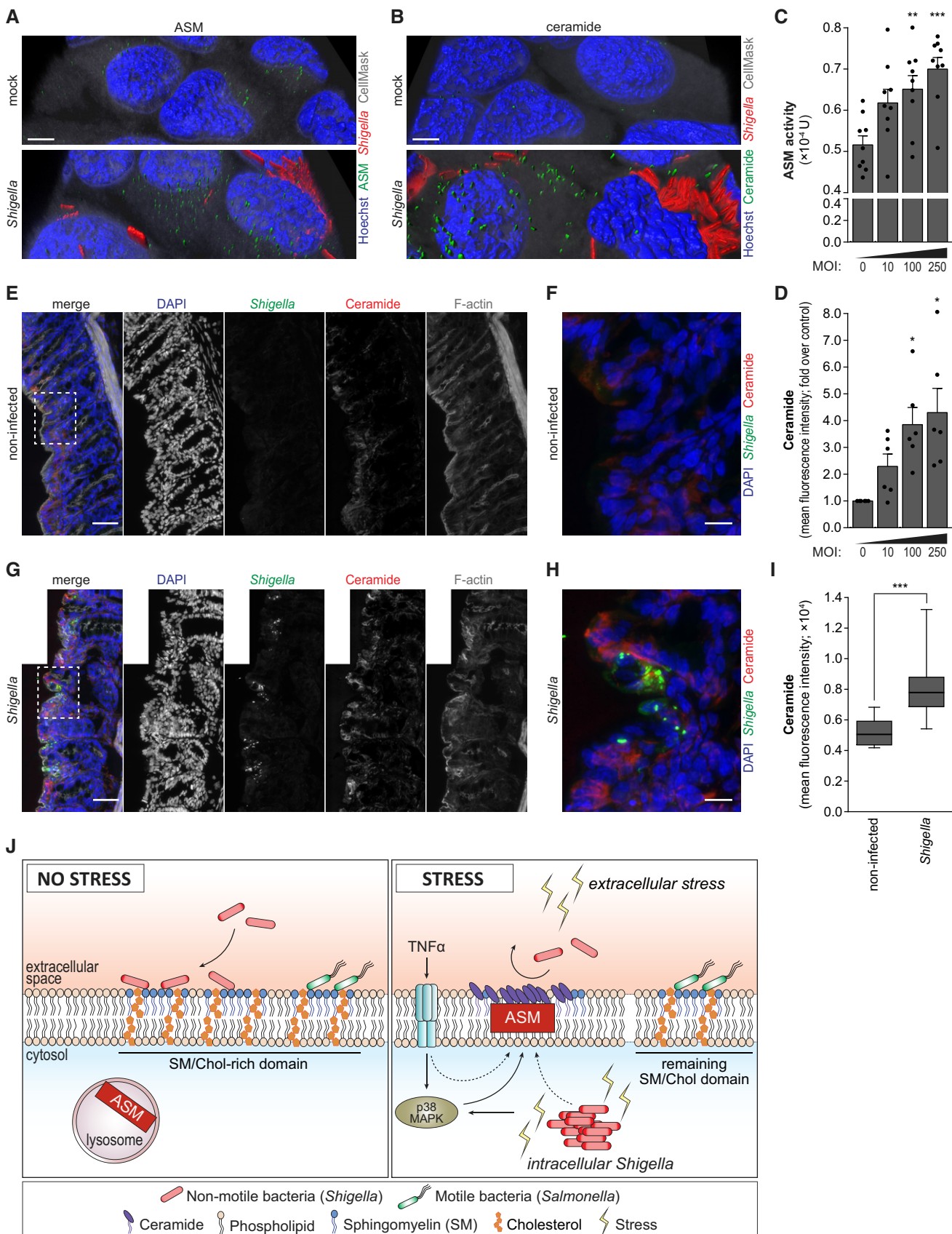

**Figure 7.**

stress protects epithelial cells from infection by non-motile pathogens might help to explain the reported impaired *in vivo* fitness of the non-motile mutant *Salmonella* strains. The relevance of motility is further reinforced by the demonstration that *L. monocytogenes* and *Y. pseudotuberculosis*, which repress flagella at 37°C and are thus non-motile when grown at this temperature (Peel *et al*, 1988; Kapatral & Minnich, 1995), also exhibit impaired binding/invasion in stress conditions.

Unlike *Salmonella*, *Shigella* has lost flagellar motility (Ewing, 1949) despite motility being an important trait for pathogenicity in the intestinal environment at various stages of infection, including reaching the sites of infection, adhesion to and invasion of host cells, and biofilm formation (reviewed in Refs: Rossez *et al*, 2015; Erhardt, 2016). However, flagellin, the major component of the bacterial flagellum, is a potent inducer of the innate immune response, and it is therefore deleterious for bacterial survival within the host, with the ensuing greater clearance of flagellated versus non-flagellated strains (Lockman & Curtiss, 1990; Dons *et al*, 2004; Lai *et al*, 2013; Olsen *et al*, 2013). This trade-off between higher motility/higher immunogenicity and lower motility/lower immunogenicity is particularly interesting in light of the results presented here.

Importantly, ASM activation and subsequent membrane remodeling under stress have propagative properties linked to the inducing stimuli. For instance, most stress conditions occurring during colonic inflammation, such as TNF-α stimulation, hypoxia, or oxidative stress, elicit responses at the level of cell populations rather than just individual cells. Moreover, a common denominator of the response to stress is the ability of cells to produce signals or activate pathways to alert neighboring cells to the presence of the stress elicitor, and orchestrate a common response (Chovatiya & Medzhitov, 2014). Along this line, our results show that, even in the context of *Shigella* infection, the remodeling of the host membrane is not restricted to infected cells, but is in fact more evident in the yet non-infected adjacent cells. Specifically, we show that *Shigella* infection induces ASM activation and strong ceramide accumulation in the bystander cells. Paracrine signaling, for example, TNF-α release upon *Shigella* infection, can explain the host membrane remodeling in the bystander cells. In addition, direct cell–cell communication can be involved, given that the p38 MAPK pathway activation has been shown to be propagated from *Shigella* infected to bystander uninfected cells via gap junctions (Kasper *et al*, 2010). These mechanisms are further supported by our results showing that stress conditions, namely TNF-α, arsenite, or anisomycin treatment, activate the p38 MAPK pathway and that this stimulation is required for the inhibition of *Shigella* infection upon stress. P38-dependent ASM activation upon infection is also in agreement with the observed stronger accumulation of ceramide in bystander cells, when compared with cells with replicating bacteria. Indeed, p38 activation in cells with internalized bacteria is modest due to the phosphothreonine-lyase activity of the *Shigella* OspF effector protein (Arbibe *et al*, 2007; Li *et al*, 2007).

We show that intracellular replication of *Shigella* induces host membrane remodeling and inhibition of re-infection by extracellular non-motile bacteria. Importantly, we have also observed membrane remodeling, visualized by a strong ceramide accumulation, in colon samples of guinea pigs infected with *Shigella*. Accordingly, we propose that membrane remodeling constitutes part of the host response to the stress elicited by *Shigella* infection, acting as a protective mechanism against further pathogen invasion. In addition

to bacterial pathogen infection, this mechanism might also be relevant to restrict commensal bacteria translocation through the intestinal epithelium. Given that the mucus layer is reduced during inflammation (Swidsinski *et al*, 2007) and thus more bacteria are probing the epithelial surface, it is conceivable that the depletion of membrane rafts limits the docking of such bacteria and their potential translocation through the intestinal barrier. Indeed, commensal bacteria have been shown to exploit membrane raft-mediated transcellular pathways to cross the intestinal epithelium, in mechanisms dependent (Kalischuk *et al*, 2009) or independent (Clark *et al*, 2005) of co-infection with bacterial pathogens.

In summary, we describe a novel mechanism by which host epithelial cell response to stress limits *Shigella* binding and invasion via reinforcement of the physical barrier formed by epithelial cells, through the activation of ASM and resulting depletion of permissive bacterial binding sites. Moreover, we provide evidence that global binding of motile pathogens to host cells is not affected in stress conditions, indicating that motility is an important feature to overcome host defenses based on host membrane remodeling.

## Materials and Methods

### Bacterial infections

For bacterial infections, overnight cultures were diluted 1:100 in LB and grown aerobically at 37°C with shaking until $OD_{600}$ 0.4 (*Shigella*) or $OD_{600}$ 2.0 (*Salmonella*). Bacteria were harvested by centrifugation (12,000 *g*, 2 min) and resuspended in complete cell culture medium. Bacterial infections were performed at the multiplicity of infection (MOI) indicated in the Figure legends. Following addition of bacteria, cells were centrifuged at 2,000 *g* for 15 min (*Shigella*) or 250 *g* for 10 min (*Salmonella*) at room temperature. Cells were then incubated at 37°C in a 5% $CO_2$ humidified atmosphere for 15 or 20 min, respectively, followed by replacement with fresh medium supplemented with 50 μg/ml gentamycin (defined as time point 0 of infection) and incubated for 30 min to kill extracellular bacteria. The medium was then replaced, and cells were maintained in medium supplemented with 10 μg/ml gentamycin, until analysis.

For binding experiments, bacteria were added to the host cells as above, and processed immediately after the centrifugation step (*Salmonella*) or after an additional 10-min incubation at 37°C (*Shigella*). Cells were then extensively washed with PBS to remove non-bound bacteria and processed for microscopy, colony-forming units (cfu) assays, or RNA extraction. For *Listeria* infections, overnight cultures were diluted 1:50 in BHI medium and grown at 37°C with shaking until $OD_{600}$ 0.7. *Listeria* was added to the cells after dilution of bacterial cultures in complete medium. Cells were then centrifuged at 250 *g* for 10 min at room temperature and incubated for 10 min at 37°C. For *Yersinia* infection, overnight culture was performed at 28°C. The culture was then diluted to $OD_{600}$ 0.05 and grown with shaking at 28°C for 1 h, followed by 2 h growth at 37°C reaching approx. $OD_{600}$ 2.0. Dilution of bacteria and infections was performed as described for *Shigella*, except that following addition of bacteria, cells were centrifuged at 400 *g* for 10 min, followed by 1-h incubation at 37°C in a 5% $CO_2$ humidified atmosphere.

To quantify intracellular bacterial replication by cfu assays, cells were washed three times with PBS and lysed in PBS containing

0.1% Triton X-100. The lysates were then serially diluted in PBS and plated on LB agar plates.

For re-infection assays, the cells were washed with PBS at 3 hpi (primary infection) and re-infected as described above. For treatment with compounds, these were added directly to the medium containing 10 μg/ml of gentamycin at the appropriate time prior to the secondary infection. Cells were then washed with PBS and re-infected. Discrimination of the colonies derived from primary and secondary infections was possible since the strains used for the re-infection experiments have different antibiotic resistance cassettes as follows: *Shigella*-mCherry, kanamycin resistance, and *Salmonella*-mCherry, ampicillin resistance (primary infection); *Shigella*-GFP and *Salmonella*-GFP (secondary infections), chloramphenicol resistance.

### Animals, infections, and sample preparation

*Shigella* infection of guinea pigs (120–250 g; Charles River Laboratories) and sample preparation were performed as described previously (Arena *et al*, 2015). Infections were performed in accordance with Institut Pasteur animal protocols (no. 2013-0113). Two animals per condition (non-infected and *Shigella* infected) were sacrificed at 4 hpi, and the distal 10 cm of colon was harvested and processed for immunofluorescence analysis.

### Immunofluorescence, fluorescence microscopy, and analysis

Cells seeded on glass coverslips were fixed with 4% paraformaldehyde (PFA) for 15 min at room temperature, followed by permeabilization with 0.5% Triton X-100 in PBS for 10 min. Blocking was performed in 1% bovine serum albumin (BSA) in PBS for 30 min. Incubation with primary antibodies was performed in blocking solution. The following primary antibodies were used: *Salmonella* LPS (1:1,000, 2 h room temperature; Abcam, ab8274), *L. monocytogenes* (1:750, 2 h room temperature; antibodies-online, ABIN237765), human ASM [1:250, overnight at 4°C; Santa Cruz, sc-293189 (4H2)], ceramide (1:50, 1 h room temperature; Enzo Life Sciences, ALX-804-196-T050). After washing with PBS, cells were incubated with the corresponding secondary antibodies conjugated with Alexa Fluor 488 or 594 (1:400, 1 h room temperature; Life Technologies).

For F-actin staining, cells were incubated with Alexa Fluor 594 Phalloidin (1:50, 1 h room temperature; Life Technologies, A12381), after the blocking step.

When indicated, cells were stained with HCS CellMask Deep Red stain (1:10,000, 1 h room temperature; Life Technologies, H32721). Nuclei were counterstained with Hoechst 33342 (1:5,000, 15 min room temperature; Life Technologies, 62249). Slides were mounted in Vectashield (Vector Labs).

Confocal microscopy images were acquired with a Leica SP5 laser scanning confocal microscope (Leica Microsystems). For the analysis of the number of bacteria per cell, at least 50 infected cells per condition and independent experiment (total of at least 250 cells) were counted manually from maximum-projected Z-stack confocal images.

The 3D reconstruction of Z-stack confocal images was performed using the Imaris software (Bitplane). An average of 20–25 Z-stacks were acquired for each image; the various channels were surface-rendered by voxel distance.

For quantification of *Shigella* infection and spreading, image acquisition was performed using Operetta automated high-content screening fluorescence microscope (PerkinElmer) at a 20× and 10× magnification, respectively; a total of 9 (infection) or 1 (spreading) images were acquired per coverslip/well, corresponding to approximately 2,500 cells analyzed. Image analysis was performed using Columbus image analysis software (PerkinElmer) as described previously (Sunkavalli *et al*, 2017).

For the guinea pig colon tissue sections (10-μm-thick transversal sections), samples were permeabilized with 0.5% Triton X-100 in PBS for 2 h and blocked in 1% BSA in PBS for 1 h. Sections were then incubated overnight at 4°C with the primary antibody against ceramide (1:50; Enzo Life Sciences, ALX-804-196-T050), a rabbit polyclonal serum specific for *Shigella*-LPS (1:300; P.S. Lab), and Alexa Fluor 647 Phalloidin (1:200; Life Technologies, A22287), in 0.1% Triton X-100/1% BSA solution. After two washes with PBS, samples were incubated for 1 h at room temperature with the goat anti-mouse and goat anti-rabbit secondary antibodies conjugated with Alexa Fluor 568 and Alexa Fluor 488 (Life Technologies), followed by DAPI staining (1:1,000; Life Technologies, 62247) for 5 min at room temperature. The sections were then washed with PBS and mounted with ProLong Gold (Life Technologies, P36930).

Images were acquired in an Opterra swept-field confocal microscope (Bruker) with a 100× objective. Montages of multiple image fields were used to reconstruct the tissue sections. The quantification of the mean fluorescent intensity (MFI) of ceramide staining in the sections was performed using Fiji (Schindelin *et al*, 2012). Briefly, maximal intensity projections of 20 z-stacks were generated and specific region of interests (ROIs) were manually drawn on the surface epithelial layer using the phalloidin staining as reference, and MFI was calculated.

### Plasma membrane acid sphingomyelinase activity assays

For determining membrane sphingomyelinase activity, HeLa cells ($3.5 \times 10^5$ cells/well) were seeded in 6-well plates 2 days prior to the assay. ASM activity was measured in the membrane fractions, as described previously (Tonnetti *et al*, 1999), using the Amplex Red Sphingomyelinase Assay Kit (Invitrogen, A12220).

To obtain the membrane fractions, following treatment with the various compounds or after infection, the cells were washed with PBS, and harvested in PBS with 5 mM EDTA and washed three times in ice-cold PBS with 100 μM sodium orthovanadate at 4°C. Cells were then resuspended in 100 μl of lysis buffer (20 mM Tris–HCl pH 7.5, 2 mM EDTA, 5 mM EGTA, 1 mM sodium orthovanadate, 10 mM β-glycerol phosphate, 1 mM PMSF, 1× protease inhibitors cocktail) and disrupted by five cycles of freezing and thawing in a methanol/dry-ice bath. The lysate was centrifuged for 10 min at 1,000 *g* at 4°C, and the supernatant (post-nuclear homogenate) was transferred to a new tube. The supernatant was centrifuged for 1 h at 100,000 *g* at 4°C. The resulting pellet (membrane fraction) was resuspended in 40 μl 1× reaction buffer provided with the kit. For each treatment/infection, a replicate well was used for cell counting; $1.2 \times 10^6$ cells were used on average as input for the extraction of the membrane fractions. 10 μl of the membrane fractions (i.e., corresponding to $3.0 \times 10^5$ cells per condition) was used for the measurement of ASM activity, which was performed according to the manufacturer's instructions. A standard curve was established for each experiment,

based on 1:4 serial dilutions of the provided commercial sphingomyelinase (10 U/ml; sphingomyelinase from *Bacillus cereus*; 1 U corresponds to the amount of enzyme required to hydrolyze 1 μmole substrate in 1 min at 37°C), starting from $4 \times 10^{-3}$ U to $0.1 \times 10^{-5}$ U in a total volume of 100 μl of 1× reaction buffer. The reactions were incubated at 37°C for 1 h, and fluorescence was measured with a microplate reader using excitation at 530 nm and emission at 590 nm. For each experiment, the fluorescence of the samples was converted to enzyme units using a standard curve.

### Ceramide quantification by flow cytometry

After treatment with stressors or infection, cells were collected in PBS + 5 mM EDTA and washed two times in PBS. Cells were then resuspended in 1% BSA in PBS containing the ceramide antibody (1:30, 1 h room temperature, with agitation; Enzo Life Sciences, ALX-804-196-T050). Cells were washed three times in PBS and again resuspended in 1% BSA in PBS containing the anti-mouse secondary antibody conjugated with Alexa Fluor 488 (1:300, 1 h room temperature, with agitation; Life Technologies). Finally, cells were washed and resuspended in cold PBS before measurement. Cells incubated with only the secondary antibody were used as a control. Flow cytometry was performed on a BD Accuri C6 or a FACSAria III flow cytometer (BD Biosciences). FlowJo software (Tree Star Inc) was used for data analysis.

### Cell viability

Analysis of cell viability was performed using 7-amino-actinomycin (7-AAD; BD Biosciences, 51-68981E), a viability dye excluded from cells with intact membranes (viable cells), as described previously (Maudet *et al*, 2014). Flow cytometry was performed on a BD Accuri C6 flow cytometer (BD Biosciences); a minimum of 10,000 cells were analyzed per sample. FlowJo software (Tree Star Inc) was used for data analysis.

### Statistical analysis

All data are presented as mean ± standard error of the mean (s.e.m.), with the exact number of experiments performed indicated in the respective Figure legend. Statistical analysis was performed using Prism software (GraphPad). For statistical comparison of datasets from two groups/conditions, two-tailed Student's *t*-test was used; for data from three or more groups/conditions, one-way ANOVA with Tukey's or Dunnett's *post hoc* test or two-way ANOVA with Sidak's multiple comparisons test was used. Mann–Whitney test was used for the comparison of medians from two groups. A *P*-value < 0.05 was considered significant. Violin plots were generated using the BoxPlotR web tool.

**Expanded View** for this article is available online.

### Acknowledgements

C.T. was a recipient of a PhD fellowship from the Graduate School of Life Sciences, University of Würzburg. This work was supported by grants from the Bavarian Ministry of Sciences, Research and the Arts in the framework of the Bavarian Molecular Biosystems Research Network (BioSysNet to J.V and A.E.), and the European Research Council (ERC) Advanced Grant (GA no. 339579 to P.S).

### Author contributions

CT and AE designed the research; CT, GN, IL, CA and CL performed the experiments; CT, GN, IL, CA, CL, MM, PS, JV, and AE analyzed the data and discussed the results; and CT and AE wrote the manuscript with input from all the authors.

### Conflict of interest

The authors declare that they have no conflict of interest.

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
