## [Review Process File · The EMBO Journal]

Stress-induced host membrane remodeling protects from infection by non-motile bacterial pathogens

Caroline Tawk, Giulia Nigro, Ines Lopes, Carmen Aguilar, Clivia Lisowski, Miguel Mano, Philippe Sansonetti, Jörg Vogel and Ana Eulalio

Review timeline:

Submission date:	30th Oct 2017
Editorial Decision:	14th Dec 2017
Revision received:	8th Jun 2018
Editorial Decision:	3rd Jul 2018
Revision received:	11th Sep 2018
Editorial Decision:	5th Oct 2018
Revision received:	5th Oct 2018
Accepted:	10th Oct 2018

Editor: Anna Leibfried / Karin Dumstrei

Transaction Report:

1st Editorial Decision

14th Dec 2017

Thank you for submitting your manuscript for consideration by the EMBO Journal. It was seen by three referees whose comments are again shown below.

As I mentioned to you before, the referees appreciate your data, but think that these do not support your conclusions sufficiently, that further reaching insight is needed, and that better controls and statistical analyses need to be performed. I had therefore asked you to provide a point-by-point response draft upfront to see how you would address the referees' concerns. Thank you for sending this draft. Based on your outline, I can consider a revised version of your manuscript. Due to the amount of new experiments to be conducted, I can extend the revision time to six months. I would like to point out that the argumentation you provided in your point-by-point response regarding bystander cells being affected due to gap junctions seems to me not very plausible in the context of your HeLa cell experiments. Furthermore, I would suggest to address referee #1's concerns about ASM activation by further supporting the proposed ROS-ASM-p38 hypothesis, perhaps by complementing the ROS measurements with a NAC treatment-based rescue experiment.

I should remind you that it is EMBO Journal policy to allow a single round of revision only and that, therefore, acceptance or rejection of the manuscript will depend on the completeness of your responses in this revised version.

REFeree REPORTS:

Referee #1:

The complex role of inflammatory stress in host defense is not fully resolved. Tawk et al investigate the stress response of epithelial cells during infection by Shigella and Salmonella, and show that stress generated by a variety of inflammatory agents (arsenite, anisomycin, hypoxia, hydrogen peroxide, cytokines) leads to host cell membrane remodeling. Although membrane remodeling inhibits Shigella invasion, Salmonella invasion is not affected due to flagella-based motility. Moreover, Shigella intracellular replication activates acid sphingomyelinase (ASM) and membrane remodeling, and inhibits Shigella re-infection. Collectively, using a variety of clever in vitro and in vivo infection models, these findings show that inflammatory stress and membrane remodeling can protect host cells against Shigella infection. Further, the authors propose that Salmonella, due to flagella-based motility, can overcome this mechanism of host defense.

These findings reveal host membrane remodeling as a cell-autonomous defense mechanism that protects epithelial cells from infection by Shigella. This is a collaborative effort involving several well established (Sansonetti, Vogel) and new (Nigro, Eulalio) labs. This report addresses several emerging and important topics in infection biology (including inflammation, bystander cells, and re-infection), working in vitro for mechanistic insight and working with guinea pigs for in vivo relevance. The quality of work is high and the messages are novel.

Major issues

1. Shigella replication can induce stress, membrane remodeling, and prevent re-infection. Can the authors provide more mechanistic insight? How does cytosolic replication activate ASM? Do changes in ASM strictly account for protection? Does the Shigella Δ IpaB/Inv strain (internalized, no replication) induce stress, ASM activation, host membrane remodeling?
2. Shigella versus Salmonella. There are many differences between these pathogens in addition to flagella-based motility. Does Salmonella intracellular replication also induce stress, ASM activation, membrane remodeling? Testing a SifA mutant (to produce cytosolic Salmonella) can also be used to test the authors' predictions.
3. Using the guinea pig model of Shigella infection, the authors test ceramide accumulation +/- infection in vivo. To test the role of membrane remodeling in host defence in vivo, can guinea pigs be pre-exposed to stress and/or re-infected? In this case, compare Shigella bacterial burdens.
4. The authors propose that Salmonella infection is not affected by membrane remodeling because they are motile. This is clearly shown in vitro but what evidence is there that flagella-based motility can matter in vivo?
5. Interestingly, membrane remodeling is not restricted to infected cells and is clearly observed in non-infected bystander cells. Considering there is no intracellular bacterial replication in bystander cells, what is the inflammatory stress / signal for bystander cells to induce membrane remodeling? Does inflammatory stress prevent intracellular cell to cell spread (IcsA-mediated)?

Minor points

The messages are clear and supported by the data. Experiments are mostly well controlled.

1. Shigella versus other 'non-motile' pathogens. If to generalize results beyond Shigella, infection using other bacterial pathogens should be tested. Instead, for this report, I suggest to keep the paper and discussion mostly focused on Shigella versus Salmonella, ie the pathogens studied herein.
2. These results show that stress can inhibit Shigella binding. Given the prominent role of IcsA in Shigella adhesion (Brotcke Zumsteg et al, Cell Host Microbe 2014), the authors can test IcsA in their model.
3. In Figure 2 the authors propose that actin dynamics are not affected, however an image showing actin-rich ruffles may not be sufficient to fully conclude this (Figure 2A). Bacterial invasion can be tested, for example does the Shigella Δ IpaB/Inv strain (internalized, no replication) invade cells equally well as wild type bacteria +/- stress?
4. Many of the compounds tested here have previously been linked to induction of host cell death pathways. Is there a role for host cell death in these results?

Referee #2:

The manuscript entitled "Stress- induced host membrane remodeling protects from infection by non-motile bacterial pathogens" by Caroline Tawk et al. describes the finding that host stress reduces binding of *Shigella flexneri* to epithelial cells. Stress regulates the acid sphingomyelinase and thereby membrane domains. Inhibition of this system neutralizes the effects of stress on adhesion of *Shigella* to host cells.

Several issues need to be addressed:

1. Why are these stress forms used to mimic inflammation. Inflammation is a complex mechanism and there are no controls that these stress forms at least mimic highlights of inflammation.
2. The concept of rafts is rather controversial and the authors need to discriminate different membrane domains.
3. Most of the bar graphs show SEM. This is not correct, since the data are not obtained from different sets of experiments. Thus, to show the true SD, most of the graph variations have to be multiplied by at least 2. I therefore doubt some of the significances, for instance in Fig. 1F (control vs H₂O₂). Further, some of the data are from 3 independent experiments only, which is insufficient, at least giving recent NIH guidelines.
4. None of the used inhibitors is specific, neither GW4869 nor amitriptyline. Amitriptyline has been shown to block the acid ceramidase in addition to the acid sphingomyelinase and without measurement of the enzyme activities in the specific set up, it is impossible to determine whether an effect is caused by acid sphingomyelinase or acid ceramidase (or both) inhibition. Further, ceramide should be quantified by mass spectrometry. Thus, the conclusions made in Fig. 3 are not conclusive.
5. The knock-down of the acid sphingomyelinase does not prove the specificity of the anti-acid sphingomyelinase antibody, because downregulation of the enzyme may block translocation of the enzyme.
Controls for the knock-down are lacking. How much is the reduction of acid sphingomyelinase activity prior or after infection? RNA levels are a hint, but are not sufficient to determine a real down-regulation of the acid sphingomyelinase. Enzyme activity measurements are required.
6. Fig. 3: It is hard to see that the Asm and ceramide are on the surface. I also do not see any membrane domains.
7. The kinase inhibitor is certainly also not specific for p38K and genetic approaches must be provided.
8. What are the cellular effects of arsenite and anisomycin? These studies are poorly controlled. Why do they change the infection? Did the authors exclude an effect on the pathogens?
9. The studies on the intracellular pathogens and the effect on e-infection are very descriptive. This applies to the in vitro as well as in vivo assays.
10. Asm activity must be given in enzyme units. Small relative increases are not informative.
11. Sphingomyelin activity should be corrected to sphingomyelinase activity. Why were the cells washed in PBS with orthovanadate? The orthovanadate might have an impact on activity of the acid sphingomyelinase.
12. The concentration of amitriptyline (50 micromolar) is extremely high. This dose is toxic.

Referee #3:

General summary:

Gastrointestinal pathogens such as *Shigella flexneri* or *Salmonella enterica* use diverse strategies to invade the intestinal epithelium of the host, which in response mounts an inflammatory response. The inflammatory environment constitutes a major stress for the intestinal epithelium, but also helps to clear invading pathogens. How the stress response of intestinal epithelial cells contributes mechanistically to the defense against pathogens remained poorly understood.

In the present manuscript, Tawk et al. demonstrate that inflammation-related stress inhibits binding of non-motile *Shigella* to epithelial cells. Various sources of stress resulted in activation of acid sphingomyelinase (ASM) and subsequent remodeling of the epithelial cell membrane, which in turn impaired the binding of non-motile *Shigella* bacteria to the host cell membrane. The authors then showed that *Salmonella* exploits flagella-mediated motility to overcome the restriction caused by stress-induced host cell membrane remodeling by accumulating at the remaining permissive entry sites.

How the host defends against invading pathogens and how bacterial pathogens evolved diverse strategies - including flagella-mediated motility - to overcome the host's defense mechanisms is of great biological significance and interest to a general audience.

The work is elegantly carried out and the manuscript is well written. The experiments mostly support the author's conclusions, but I have the following comments and suggestions to improve the author's conclusions and the clarity of the manuscript.

Major comments:

1) Induction of cellular stress using sodium arsenite treatment appears to be cell-type specific. Why would infection of HeLa cells be more affected upon sodium arsenite than HCT-8 cells? It would be useful to demonstrate that the inhibition of *Shigella* infection upon stress induction is a general effect and not specific to a few cell lines.

2) It is unclear why inhibition of protein synthesis by puromycin and cycloheximide did not affect *Shigella* infection (Fig. S1C-E), but inhibition of protein synthesis by anisomycin (Fig. 1E-F) did? Further, the effect of stress induced by anisomycin on *Shigella* infection was overcome by inhibiting ASM activity. However, how does the cell make ASM in the first place in the presence of a protein synthesis inhibitor? Is ASM present (but inactive?) before treatment with anisomycin? This would contradict with the results shown in Fig. S4 i.e. that the activation of the p38 MAPK pathway by arsenite and anisomycin is required for ASM activation.

3) The effect of cellular stress induction on *Salmonella* infection is inconsistent. The authors state that upon infection with *Salmonella*, fewer cells were infected, but more bacteria adhered per cell (based on fluorescence microscopy analyses; Fig. 4A, D-F). However, the number of intracellular *Salmonella* (based on CFU counts and *Salmonella* mRNA quantification) were not affected (Fig. 4B-C)? Would it be possible that *Salmonella* replicates more efficiently inside arsenite/anisomycin-stressed cells?

4) The authors demonstrate that the infection with *Shigella* induces ASM activation and ceramide accumulation also in bystander cells. This suggests a general epithelial cell defense mechanism, where bystander cells next to infected cells would be immune to infection, i.e. ceramide-positive cells should not be infected by *Shigella*, which might be easily tested using e.g. flow cytometry analysis.

Minor comments:

1) The fluorescence images in Figs 1B, 1C, 1E, 2A, 2D etc. would benefit from quantification.

Compared to the shown CFU data, the impaired Shigella infection is less convincing in the presented fluorescence images.

2) The swimming behavior of Salmonella near the surface of host cells has been extensively studied before and should be cited (PMID 22911370) - page 8, top.

3) The use of a Δ fliC Salmonella mutant as a non-motile strain is curious. As noted by the authors, Salmonella harbors two distinct flagellins (FliC and FljB), whose expression is stochastic. Thus, a subset of a population of Δ fliC mutant Salmonella would still be motile (which is also noted by the authors).

However, the frequency of switching flagellin expression is low (around 1/5000 per cell per generation) and biased towards switching to the fliC_{ON} orientation (PMID 1848842). Accordingly, a population of Δ fliC mutant Salmonella would primarily be non-motile and the infection phenotype similar to a complete flagella mutant (Δ flhC), which is consistent with the author's results shown in Fig. 4G-H. This should be noted in the manuscript and the rationale of using both a Δ fliC and Δ flhC mutant should be explained.

4) Is the ability of Salmonella to accumulate at the remaining permissive entry sites dependent on the expressed flagellin? It has been shown that the near-surface swimming behavior of Salmonella is dependent on which flagellin (FliC or FljB) is expressed (PMID 28295924), which might fit with the author's observations.

5) Another interesting experiment would be to mix a population of non-motile and motile Salmonella bacteria (labelled with different fluorescence proteins or different antibiotic resistance markers) and perform a co-infection experiment. The author's model would predict that only motile bacteria would be able to infect and thus out-compete a non-motile Salmonella mutant. Such an experiment would nicely complement the author's Shigella-Salmonella co-infection experiment while circumventing any inter-species effects.

In this respect, I would recommend to use a Salmonella mutant of the motor-force generators (MotAB), which impairs rotation (and thus motility function) of the flagellum, but does not affect the role of flagella in binding to surfaces.

6) Does a primary infection with Salmonella induce host membrane remodeling and prevent a secondary infection with Shigella?

7) The bacterial gene nomenclature throughout the manuscript should be verified (i.e. the correct nomenclature of gene deletions is lowercase and italics, e.g. Δ fliC)

Additional suggestions:

1) Fig. 5C; 5E: How were Shigella colonies arising from the primary and secondary infection determined (also, how were Shigella and Salmonella colonies discriminated)? Did the authors use different antibiotic resistance cassettes? If so, this should be mentioned in the Methods section.

2) Discussion: Temperature-sensitive flagella expression of Listeria and Yersinia. My general understanding is that during the initial colonization of the gut, the pathogens are still motile, since infection of the epithelium occurs rapidly after ingestion of the pathogens (which expressed flagella outside the host).

The temperature-dependent downregulation of flagella expression would only affect de novo synthesis of flagella and might be more important for intracellular survival and systemic infection?

3) Page 11: rephrase '...leading to inhibition Salmonella invasion...'

'Stress-induced host membrane remodeling protects from infection by non-motile bacterial pathogens'

Referee #1:

The complex role of inflammatory stress in host defense is not fully resolved. Tawk et al investigate the stress response of epithelial cells during infection by *Shigella* and *Salmonella*, and show that stress generated by a variety of inflammatory agents (arsenite, anisomycin, hypoxia, hydrogen peroxide, cytokines) leads to host cell membrane remodeling. Although membrane remodeling inhibits *Shigella* invasion, *Salmonella* invasion is not affected due to flagella-based motility. Moreover, *Shigella* intracellular replication activates acid sphingomyelinase (ASM) and membrane remodeling, and inhibits *Shigella* re-infection. Collectively, using a variety of clever in vitro and in vivo infection models, these findings show that inflammatory stress and membrane remodeling can protect host cells against *Shigella* infection. Further, the authors propose that *Salmonella*, due to flagella-based motility, can overcome this mechanism of host defense.

These findings reveal host membrane remodeling as a cell-autonomous defense mechanism that protects epithelial cells from infection by *Shigella*. This is a collaborative effort involving several well established (Sansonetti, Vogel) and new (Nigro, Eulalio) labs. This report addresses several emerging and important topics in infection biology (including inflammation, bystander cells, and re-infection), working in vitro for mechanistic insight and working with guinea pigs for in vivo relevance. The quality of work is high and the messages are novel.

Reply: We are very grateful to the Referee for her/his supportive evaluation of our work and insightful comments.

Major issues

1. *Shigella* replication can induce stress, membrane remodeling, and prevent re-infection. Can the authors provide more mechanistic insight? How does cytosolic replication activate ASM? Do changes in ASM strictly account for protection? Does the *Shigella* Δ pab/Inv strain (internalized, no replication) induce stress, ASM activation, host membrane remodeling?

Reply: Concerning the mechanisms underlying stress and membrane remodeling induced by *Shigella*, our results indicate that the activation of p38 MAPK pathway upon *Shigella* infection is the main mechanism leading to membrane remodeling and preventing reinfection. Indeed our new data obtained in reinfection experiments (Fig. 6L, 6M, EV5D, EV5E in the revised manuscript) show that upon experimental inhibition of the p38 pathway, either by chemical inhibition (using SB203580) or p38 knockdown (using siRNAs), the secondary infection by *Shigella*-GFP is not inhibited by the primary infection. The relevance of p38 pathway in the context of stress is further supported by the results added to the revised version of the manuscript showing that the various stress conditions used in the manuscript, namely treatment with arsenite, anisomycin, TNF- α and H₂O₂, all converge in the activation of the p38 MAPK pathway (Fig. 4A). Accordingly, inhibition of the p38 pathway (SB203580 treatment and new data following p38 knockdown) also blunts the inhibitory effect of arsenite or anisomycin on *Shigella* infection (Fig. 4B-F, EV3A-C).

Our data supports indeed the notion that in the context of infection, ASM activation and consequent membrane remodeling accounts for the observed protection against re-infection. The evidence supporting this conclusion is the nearly complete recovery of infection levels in conditions of ASM inhibition, either by treatment with the ASM inhibitor amitriptyline or following ASM knockdown by siRNA (Fig. 6I-K, EV5A-C). However, we cannot exclude contributions of other pathways/mechanisms to the observed phenotype.

As requested by the Referee, we have tested if the *Shigella* $\Delta ipaB/Inv$ strain (which invades mammalian cells, but is unable to disrupt the vacuole and replicate) leads to ASM activation and ceramide production. As shown in Fig. EV5R and EV5S of the revised manuscript, this strain did not lead to membrane remodeling, which reinforces the notion that cytoplasmic *Shigella* replication is required for ASM activation and membrane remodeling and is in full agreement with the observation that primary infection with this mutant strain does not compromise re-infection.

2. *Shigella* versus *Salmonella*. There are many differences between these pathogens in addition to flagella-based motility. Does *Salmonella* intracellular replication also induce stress, ASM activation, membrane remodeling? Testing a SifA mutant (to produce cytosolic *Salmonella*) can also be used to test the authors' predictions.

Reply: We fully agree with the Referee that, in addition to flagellar motility, there are other important differences between *Shigella* and *Salmonella*. Importantly, the disparate intracellular lifestyles of these bacterial pathogens, with *Shigella* replication occurring in the host cytoplasm whereas that of *Salmonella* being, to a large extent, confined to a vacuole, might result in different phenotypes regarding ASM activation and consequent membrane remodeling. To address this point we have followed the suggestion of the Referee and evaluated the ability of *Salmonella* WT and the $\Delta sifA$ mutant (which escapes the *Salmonella* containing vacuole and replicates in the host cytosol) to induce ceramide production. We did not observe an increase of ceramide in the plasma membrane following infection with *Salmonella* WT or $\Delta sifA$ (Fig. S3F), in infected or bystander cells, indicating that bacterial cytoplasmic replication is not, per se, responsible for the membrane remodeling. In the case of *Salmonella* WT the experiments were performed at 20 hpi, a time-point at which there is extensive intracellular replication; for the $\Delta sifA$ mutant, experiments were performed at 6 hpi, a time-point at which cytosolic replication is readily detected. Of note, these experiments were performed in HCT-8 cells and not in HeLa-229 cells, since we have observed that in the latter cell type *Salmonella* is not able to efficiently replicate in the cytosol. This is in contrast with HeLa-CCL2, where we have observed cytosolic replication for *Salmonella* WT and the $\Delta sifA$ mutant, in agreement with the literature (Beuzon et al, 2002; Malik-Kale et al, 2012).

3. Using the guinea pig model of *Shigella* infection, the authors test ceramide accumulation +/- infection in vivo. To test the role of membrane remodeling in host defence in vivo, can guinea pigs be pre-exposed to stress and/or re-infected? In this case, compare *Shigella* bacterial burdens.

Reply: To address the Referee's point on the relevance of membrane remodeling to the host defense in vivo, we have attempted to establish a *Shigella* re-infection model in guinea pig, which is an experiment that has not been performed before (please note that approval of stress induction by arsenite in guinea pig would take too long for it to be obtained, if at all, within the timeframe of our manuscript revision).

Initially, we predicted that the most challenging aspect of these experiments would be related to the timing of the primary and secondary infections, in order to achieve a balance between efficient membrane remodeling, while avoiding excessive colonic tissue damage induced by the primary infection. However, other technical issues prevented the success of these experiments. The experimental set-up was designed so that primary infection was performed for 3 hours, a time at which the secondary infection was performed. However, the primary infection was consistently not efficient, with little *Shigella* epithelial intracellular replication and high bacterial accumulation in the intestinal lumen. We suggest several possible explanations for these observations including: i) the guinea pigs had to be anaesthetized twice, which could impact the intestinal motility and, consequently, bacterial infection; ii) the intrarectal inoculation of the bacteria and associated pressure, also performed twice, could also affect the bacterial infection; iii) the use of a new batch of guinea pigs, with potentially different intestinal microbiota, might have impacted *Shigella* infection.

4. The authors propose that *Salmonella* infection is not affected by membrane remodeling because they are motile. This is clearly shown in vitro but what evidence is there that flagella-based motility can matter in vivo?

Reply: Evidence that *Salmonella* flagellar-based motility matters in vivo indeed exists and we agree with the Referee that this is a very pertinent point that should had been properly addressed in the initial version of the manuscript. This point was clarified in the Discussion section revised version of the manuscript.

The group of WD Hardt has elegantly shown that mutations affecting *Salmonella* motility impair bacterial colonization of the inflamed mouse intestine, but not of the normal gut (Stecher et al, 2008; Stecher et al, 2004). This is, at least in part, attributed to flagella facilitated chemotactic movement, which facilitates access to the nutrient-rich molecules released as part of the mucosal defense during inflammation (Stecher et al, 2008).

The data presented in this manuscript showing that, under stress conditions mimicking the inflamed environment, bacterial motility is important for efficient interaction with the host cell surface, is in agreement with the in vivo observations reported above. Moreover, our findings showing that host membrane remodeling in response to stress is part of the cell-autonomous defense strategies to protect epithelial cells from infection by non-motile pathogens might contribute to explain the impaired in vivo fitness of the non-motile mutant *Salmonella* strains reported by the Hardt laboratory. A discussion of this is now included in the revised manuscript (page 13).

5. Interestingly, membrane remodeling is not restricted to infected cells and is clearly observed in non-infected bystander cells. Considering there is no intracellular bacterial replication in bystander cells, what is the inflammatory stress / signal for bystander cells to induce membrane remodeling? Does inflammatory stress prevent intracellular cell to cell spread (IcsA-mediated)?

Reply: We fully agree with the Referee that the observation that membrane remodeling occurs in bystander cells is interesting. In fact, both ASM and ceramide accumulated to a higher extent in bystander cells, when compared to cells with intracellular bacteria (Fig. 7A, 7B, EV5O-Q). We hypothesize that there are two concurrent mechanisms promoting membrane remodeling in *Shigella* infected and bystander cells: 1) paracrine signaling via TNF- α release upon *Shigella* infection; and 2) direct cell-cell communication, given that the p38 MAPK pathway activation has been shown to be

propagated from *Shigella* infected to bystander uninfected cells via gap-junctions (Kasper et al, 2010). Of note, although it is usually described that HeLa cells do not express connexin 43 and consequently do not form gap-junctions, the HeLa-229 cells used in our experiments (HeLa CCL-2.1, ATCC) do indeed express high levels of connexin 43 (see Western blot below), in agreement with what has been reported in the literature (King et al, 2000). These points have been discussed in the revised version of the manuscript (page 14).

Concerning our observations of stronger accumulation of ceramide in bystander cells, when compared with cells with replicating bacteria, this is likely related to the phosphothreonine-lyase activity of the *Shigella* OspF effector protein (Arbibe et al, 2007; Li et al, 2007), which blunts p38 MAPK activation in cells with internalized bacteria. In agreement with the relevance of p38 MAPK pathway activation, we showed that when the p38 pathway is inhibited (using the inhibitor SB203580 or knockdown of p38) the secondary infection by *Shigella*-GFP was not inhibited by the primary infection (Fig. 6L, 6M, EV5D, EV5E of the revised manuscript).

To assess whether stress affects *Shigella* cell-to-cell spreading, we have determined the extent of *Shigella* spreading in control and arsenite or anisomycin treated cells, by quantifying the area of *Shigella* infection foci using fluorescence microscopy and automated image analysis, an approach we have successfully applied previously (Sunkavalli et al, 2017). This analysis revealed that the average foci size was comparable in arsenite or anisomycin treated cells and in the corresponding controls (Fig. S1J of the revised manuscript), thus excluding an effect of stress in the actin based spreading of *Shigella* to neighboring cells. Of note, infection of cells treated with arsenite or anisomycin was performed with a higher MOI than that used for control cells [MOI 100 (arsenite), 50 (anisomycin) and 10 (control)], to achieve a comparable level of bacterial invasion.

Minor points

The messages are clear and supported by the data. Experiments are mostly well controlled.

1. *Shigella* versus other 'non-motile' pathogens. If to generalize results beyond *Shigella*, infection using other bacterial pathogens should be tested. Instead, for this report, I suggest to keep the paper and discussion mostly focused on *Shigella* versus *Salmonella*, ie the pathogens studied herein.

Reply: We consider that it would be very interesting to evaluate whether stress induced membrane remodeling also impairs infection by other non-motile bacterial pathogens. As referred in the manuscript, relevant candidates include *Listeria monocytogenes* and *Yersinia pseudotuberculosis*, which repress flagella at 37°C. Motivated by the comment of the Referee, we performed additional sets of experiments using these two bacteria. Our results clearly demonstrate that, in addition to *Shigella*, arsenite treatment impairs infection by *Listeria* and *Yersinia*, when grown in conditions that repress flagella production (Fig. 5I-L of the revised manuscript). This data was added to the manuscript to further reinforce the relevance of stress induced membrane remodeling as a cell-autonomous defense mechanism that protects epithelial cells from infection by non-motile bacterial pathogens including, but not restricted to, *Shigella*.

2. These results show that stress can inhibit *Shigella* binding. Given the prominent role of IcsA in *Shigella* adhesion (Brotcke Zumsteg et al, Cell Host Microbe 2014), the authors can test IcsA in their model.

Reply: As suggested by the Referee, we have tested whether stress induced by arsenite or anisomycin affects binding of the *Shigella* Δ icsA strain to host cells. The inhibitory effect of both treatments was comparable for the *Shigella* Δ icsA, Δ ipaB and WT strains (Figs. S1H, S1I and 2A-C of the revised manuscript). These results indicate that in the experimental conditions used the effect of the stressors is independent of the IcsA role in *Shigella* adhesion.

3. In Figure 2 the authors propose that actin dynamics are not affected, however an image showing actin-rich ruffles may not be sufficient to fully conclude this (Figure 2A). Bacterial invasion can be tested, for example does the *Shigella* Δ ipaB/Inv strain (internalized, no replication) invade cells equally well as wild type bacteria +/- stress?

Reply: Regarding the invasion with *Shigella* Δ ipaB/Inv strain in cells treated with stressors, we have performed this experiment but concluded that it might not be very informative. We observed that arsenite pre-treatment significantly impairs *Shigella* Δ ipaB/Inv strain invasion, albeit to a lesser extent than the WT bacteria (see graph below). Given that *Yersinia* invasin protein binds to β 1-integrin and that it has been reported that endocytosis of β 1-integrin is lipid-raft mediated (Vassilieva et al, 2008), the inhibitory effect of arsenite is not surprising. We can only speculate that this effect being weaker for *Shigella* WT is related to the extent of the dependence on the membrane domains. Of note, we also observed that arsenite pre-treatment affects *Yersinia pseudotuberculosis* invasion (dependent on the invasin; Fig. 5K and 5L of the revised manuscript), albeit to a lesser extent than *Shigella*.

Alternatively, to demonstrate that actin dynamics impairment is not the limiting step in stress conditions we have quantified the efficiency of ruffle formation, i.e. the percentage of ruffles formed upon bacterial contact, in cells treated with stressors compared to control cells. We have not detected significant differences between cells treated with the stressors and control cells (Fig. S1G of the revised manuscript). This result further supports the notion that in stress conditions the limiting step is bacterial adhesion (supported by the *Shigella* Δ ipaB experiments, Fig. 2A-E, S1B, S1C S1E and S1F), and not the invasion step (including ruffle formation).

4. Many of the compounds tested here have previously been linked to induction of host cell death pathways. Is there a role for host cell death in these results?

Reply: We have assessed cell viability by flow cytometry, using 7-Amino-Actinomycin (7-AAD), a viability dye excluded from cells with intact membranes (viable cells), and have not detected significant effects of the various compounds on host cell viability, at the concentrations and incubation periods tested. These results have been introduced in the revised version of the manuscript (Fig. EV1J).

Referee #2:

The manuscript entitled "Stress- induced host membrane remodeling protects from infection by non-motile bacterial pathogens" by Caroline Tawk et al. describes the finding that host stress reduces binding of *Shigella flexneri* binding to epithelial cells. Stress regulates the acid sphingomyelinase and thereby membrane domains. Inhibition of this system neutralizes the effects of stress on adhesion of *Shigella* to host cells.

Several issues need to be addressed:

1. Why are these stress forms are used to mimic inflammation. Inflammation is a complex mechanisms and there are no controls that these stress forms at least mimic highlights of inflammation.

Reply: We would like to thank the Referee for her/his valuable comments and suggestions to improve our manuscript.

All the stress inducers used in the manuscript have been widely used to mimic the stress conditions encountered by cells during inflammation (arsenite, anisomycin) or are per se stimuli/cues encountered during inflammation (hypoxia, TNF- α and H₂O₂), as described in the literature. All of these stimuli converge in the production of reactive oxygen species (ROS), key signaling molecules during inflammation. Of note, arsenite and anisomycin are well described to cause oxidative stress through the production of ROS [e.g. (Chambers & LoGrasso, 2011; Kumagai & Sumi, 2007; Liu et al, 2001; Ruiz-Ramos et al, 2009; Torocsik & Szeberenyi, 2000)]. This information has been reinforced in the revised version of the manuscript. The concentrations and duration of treatments used for the different compounds also conform to what has been described in the literature. Experiments showing that the treatment with the antioxidant N-acetylcysteine (NAC) reverts the inhibitory effect of arsenite or anisomycin on *Shigella* infection further confirm that the oxidative stress induced upon treatment with these stressors is mediating their effect on *Shigella* infection (Fig. 1G and 1H of the revised manuscript).

Moreover, as described in point 1 of Referee 1, we have determined that the various stress conditions used in the manuscript, namely treatment with arsenite, anisomycin, TNF- α and H₂O₂, activate the p38 MAPK pathway (Fig. 4A of the revised manuscript), a well described player in inflammation [reviewed in (Schieven, 2005; Yang et al, 2014)]. Accordingly, p38 pathway inhibition (by knockdown of p38 or chemical inhibition with SB203580 treatment) blunted the inhibition of the *Shigella* secondary infection in re-infection assays (Fig. 6L, 6M, EV5D, EV5E of the revised manuscript) and reverted the inhibitory effect of arsenite or anisomycin on *Shigella* infection (Fig. 4B-F, EV3A-C of the revised manuscript).

2. The concept of rafts is rather controversial and the authors need to discriminate different membrane domains.

Reply: We agree with the Referee that the concept of lipid rafts remains a subject of debate. Importantly, recent studies have supported key roles for these domains in various cellular functions,

justifying their conservation throughout evolution [recently reviewed in (Sezgin et al, 2017)]. The role of membrane domains/lipid rafts has also been well described in the context of host-pathogen interactions [reviewed for example in (Bagam et al, 2017; Lafont & van der Goot, 2005)]. Along this line, our manuscript clearly demonstrates that the stress induced host membrane remodeling via activation of the acid sphingomyelinase and the generation of ceramide domains has an impact on infection by non-motile pathogens. According to the Referee's request, we have clarified where appropriate that we are referring to ceramide domains.

3. Most of the bar graphs show SEM. This is not correct, since the data are not obtained from different sets of experiments. Thus, to show the true SD, most of the graph variations have to be multiplied by at least 2. I therefore doubt some of the significances, for instance in Fig. 1F (control vs H₂O₂). Further, some of the data are from 3 independent experiments only, which is insufficient, at least giving recent NIH guidelines.

Reply: The results presented in the original manuscript were obtained from independent experiments (in most cases at least 5 independent experiments, see below) and our objective is to compare the means from different groups to determine if they are different, therefore it is our understanding that it is correct to use SEM (e.g. (Cumming et al, 2007; Sullivan et al, 2016) and EMBO Journal Author Guidelines). Therefore, we propose to maintain the graphs with SEM in the bar graphs, but we will follow any other indication deemed appropriate by the Editorial team. The number of independent experiments is clearly stated in all figure legends.

In any case, it should be stressed that statistical significance is obviously independent of the way in which the data is presented (SEM or SD) and therefore no doubts should be raised concerning this matter. In the example selected by the Referee (Figure 1F, control vs. H₂O₂, p-value = 0.0015).

As mentioned above, the large majority of the data presented in the original version of the manuscript derived from at least 5 independent experiments. In the two cases where only 3 experiments were performed (corresponding to Fig. 5E and 5F), we have performed additional experiments to achieve at least 5 independent experiments throughout the manuscript. Of note, all the new results included in the revised version of the manuscript have also been obtained from at least 5 independent experiments.

4. None of the used inhibitors is specific, neither GW4869 nor amitriptyline. Amitriptyline has been shown to block the acid ceramidase in addition to the acid sphingomyelinase and without measurement of the enzyme activities in the specific set up, it is impossible to determine whether an effect is caused by acid sphingomyelinase or acid ceramidase (or both) inhibition. Further, ceramide should be quantified by mass spectrometry. Thus, the conclusions made in Fig. 3 are not conclusive.

Reply: GW4869 and amitriptyline are widely used as inhibitors of neutral sphingomyelinase (NSM) and acid sphingomyelinase (ASM), respectively [for example (Awojodu et al, 2014; Faulstich et al, 2015; Grimm et al, 2005; Luberto et al, 2002; Luisoni et al, 2015)]. While GW4869 did not rescue the impairment of *Shigella* infection prompted by arsenite or anisomycin (Fig. EV2A-F), amitriptyline reverted the inhibitory effect of the two stress inducers (Fig. 3A-F), implicating ASM in the process. Being aware of the limitations of using chemical inhibitors, we complemented these observations by performing knockdown of ASM using siRNAs (Fig. 3G-I). Knockdown of ASM replicated the phenotype observed with amitriptyline, thus excluding a potential effect of acid ceramidase in the process.

Notwithstanding, we have validated the amitriptyline effect in our experimental set-up by measuring ASM enzymatic activity as requested by the Referee (Fig. EV2K of the revised manuscript). Moreover, we have quantified ceramide production. Instead of mass-spectrometry, as suggested by the Referee, we applied a flow-cytometry based approach (see also point 6 of Referee 2), which is a simpler, also well described method for ceramide quantification [for example (Grassme et al, 2003; Lu et al, 2012; Simonis et al, 2014)]. These results have been introduced in the revised version of the manuscript (Fig. EV2L, EV2M and 7D).

5. The knock-down of the acid sphingomyelinase does not prove the specificity of the anti-acid sphingomyelinase antibody, because downregulation of the enzyme may block translocation of the enzyme.

Controls for the knock-down are lacking. How much is the reduction of acid sphingomyelinase activity prior or after infection? RNA levels are a hint, but are not sufficient to determine a real down-regulation of the acid sphingomyelinase. Enzyme activity measurements are required.

Reply: As suggested by the Referee, we have measured ASM activity in our experimental conditions to confirm the ASM knockdown efficiency (Fig. EV2P). Of note, we have also performed Western-blot with the ASM antibody and confirmed the reduction of the protein levels. Although this data was not included in the initial submission, we have included it in the revised version of the manuscript (Fig. EV2O).

6. Fig. 3: It is hard to see that the Asm and ceramide are on the surface. I also do not see any membrane domains.

Reply: We concede that microscopy alone does not conclusively demonstrate the localization of ASM and ceramide at cell surface. To address this point, we have quantified ceramide by flow-cytometry, using live cells for the labelling. In live cells, the ceramide antibody can only recognize extracellular/exposed epitopes and thus exclusively labels ceramide present at the cellular surface. This analysis revealed a significant increase of ceramide in cells treated with the stressors arsenite and anisomycin (prevented by amitriptyline treatment), as well as in *Shigella* infected cells. We are confident that this data unequivocally demonstrates the ASM relocation upon stress, and the presence of ceramide at cell surface. This data was introduced in the revised version of the manuscript as Fig. EV2L, EV2M and 7D.

Please note that we do not claim that we visualized the accumulation of ASM or ceramide in membrane domains. Indeed, the lipid-rafts/membrane domains are described as nanoscopic domains [<200 nm; (Pralle et al, 2000; Varma & Mayor, 1998)], and therefore cannot be resolved by conventional optical microscopy.

7. The kinase inhibitor is certainly also not specific for p38K and genetic approaches must be provided.

Reply: To further strengthen the experiments shown in the original manuscript demonstrating that the inhibition of *Shigella* infection during stress depends on the activation of the p38 MAPK pathway, we have repeated the experiments in p38 knockdown cells (using p38 siRNA), as requested by the Referee. In agreement with the chemical inhibitor results, in p38 knockdown cells the inhibitory effect of arsenite or anisomycin on *Shigella* infection is dampened or completely abolished (Fig. 4D, 4F, EV3A and EV3C of the revised manuscript). Moreover, the p38 knockdown blunted the inhibition of the secondary infection in *Shigella* re-infection assays (Fig. 6L and EV5D of the revised manuscript).

8. What are the cellular effects of arsenite and anisomycin? These studies are poorly controlled. Why do the change the infection? Did the authors exclude an effect on the pathogens?

Reply: As referred in point 1 of Referee 2 comments, arsenite and anisomycin are widely used stressors described to cause oxidative stress through the production of reactive oxygen species [ROS; e.g. (Chambers & LoGrasso, 2011; Kumagai & Sumi, 2007; Liu et al, 2001; Ruiz-Ramos et al, 2009; Torocsik & Szeberenyi, 2000)]. As described in point 1, experiments performed with the antioxidant N-acetylcysteine (NAC) reverted the inhibitory effect of arsenite or anisomycin on *Shigella* infection, further confirm that the oxidative stress induced upon treatment with these stressors is mediating their effect on *Shigella* infection (Fig. 1G and 1H of the revised manuscript). Moreover, we have also shown that both stressors, as well as others used in the manuscript, lead to p38 MAPK pathway activation (Fig. 4A of the revised manuscript).

Regarding the effect of these compounds on the pathogens, Fig. S1B of the original manuscript showed that arsenite has no effect on *Shigella* growth when added directly to the bacterial growth medium. We have extended these experiments to also check the effect of anisomycin on the growth of both *Shigella* and *Salmonella* and of arsenite on *Salmonella* growth. In the case of *Shigella*, neither arsenite nor anisomycin treatment affected bacterial growth. For *Salmonella*, anisomycin did not affect bacterial growth, whereas arsenite significantly reduced growth when added directly to the LB medium. However, it should be noted that the experimental design is such that the stressors are not added directly to bacteria but instead are added exclusively to host cells, which are extensively washed prior to addition of bacteria. Indeed, *Salmonella* infection is not affected by the arsenite pre-treatment (in terms of total CFU counts). We have included these data in the revised version of the manuscript (Fig. EV1K). To further exclude the effect of arsenite on the intracellular replication of *Salmonella*, we have also shown that arsenite treatment performed post-invasion (0.5 h post-infection, for 1 h) did not affect bacterial replication, as evaluated by CFU assay at 20 hpi (Fig. S3B and S3C; see also below Referee 3, point 3).

9. The studies on the intracellular pathogens and the effect on e-infection are very descriptive. This applies to the in vitro as well as in vivo assays.

Reply: We do not agree with this comment of the Referee. In fact, throughout the manuscript we have made a significant effort to complement all observations with relevant experiments aimed at investigating the phenotypes in detail and at drawing causal relationships concerning the underlying mechanisms (e.g. chemical and genetic approaches to inhibit key potential molecular players, use of mutant strains, re-infection with different bacteria +/- ASM or p38 inhibition and siRNAs). We are confident that the additional experiments motivated by the comments of the three Referees have further strengthened the manuscript.

10. Asm activity must be given in enzyme units. Small relative increases are not informative.

Reply: As requested by the Referee, we have calculated ASM activity in enzyme units by performing a standard curve for calibration (described in Materials and Methods section). The ASM activity in mU/h is included in the revised version of the manuscript, indicated in the figure legends for the 100% activity (Fig. EV2K, EV2P, 7C). Given that ASM activity is frequently shown normalized to control conditions [for example (Bianco et al, 2009; Edelmann et al, 2011; Gorelik et al, 2016; Simonis et al,

2014)], and that it is easier to have a perception of the changes occurring in comparison with other studies, we decided to maintain the data representation as relative changes.

11. Sphingomyelin activity should be corrected to sphingomyelinase activity. Why were the cells washed in PBS with orthovanadate? The orthovanadate might have an impact on activity of the acid sphingomyelinase.

Reply: Acid sphingomyelinase activity was measured in the membrane fractions isolated using a protocol previously described by Tonnetti et al. (Tonnetti et al, 1999), that similarly to other protocols [e.g. (Grassme et al, 2005)], includes orthovanadate in the lysis buffer. Moreover, we have confirmed by enzymatic assays that orthovanadate does not affect ASM activity.

We have corrected “sphingomyelin activity” to “sphingomyelinase activity”.

12. The concentration of amitriptyline (50 micromolar) is extremely high. This dose is toxic.

Reply: In our experiments we have used 50 μM amitriptyline for a maximum incubation period of 75 minutes. Similar use of this compound has been reported in the literature without affecting cell viability [e.g. (Awojoodu et al, 2014; Petersen et al, 2013)]. Lower concentrations (e.g. 10 μM or 25 μM) can also be found in the literature, but these were typically used for significantly longer treatment periods [e.g. (Luisoni et al, 2015)].

We have confirmed that indeed 50 μM amitriptyline is not toxic in our experimental setup (Fig. EV1J of the revised manuscript), by accessing cell viability by flow cytometry following labelling with 7-Amino-Actinomycin (7-AAD), a viability dye excluded from cells with intact membranes (viable cells).

Referee #3:

General summary:

Gastrointestinal pathogens such as *Shigella flexneri* or *Salmonella enterica* use diverse strategies to invade the intestinal epithelium of the host, which in response mounts an inflammatory response. The inflammatory environment constitutes a major stress for the intestinal epithelium, but also helps to clear invading pathogens. How the stress response of intestinal epithelial cells contributes mechanistically to the defense against pathogens remained poorly understood.

In the present manuscript, Tawk et al. demonstrate that inflammation-related stress inhibits binding of non-motile *Shigella* to epithelial cells. Various sources of stress resulted in activation of acid sphingomyelinase (ASM) and subsequent remodeling of the epithelial cell membrane, which in turn impaired the binding of non-motile *Shigella* bacteria to the host cell membrane. The authors then showed that *Salmonella* exploits flagella-mediated motility to overcome the restriction caused by stress-induced host cell membrane remodeling by accumulating at the remaining permissive entry sites.

How the host defends against invading pathogens and how bacterial pathogens evolved diverse strategies - including flagella-mediated motility - to overcome the host's defense mechanisms is of great biological significance and interest to a general audience.

The work is elegantly carried out and the manuscript is well written. The experiments mostly support the author's conclusions, but I have the following comments and suggestions to improve the author's conclusions and the clarity of the manuscript.

Reply: We are very grateful to the Referee for her/his careful evaluation of our work and for the insightful and constructive comments.

Major comments:

1) Induction of cellular stress using sodium arsenite treatment appears to be cell-type specific. Why would infection of HeLa cells be more affected upon sodium arsenite than HCT-8 cells? It would be useful to demonstrate that the inhibition of *Shigella* infection upon stress induction is a general effect and not specific to a few cell lines.

Reply: We fully agree with the Referee that it is important to verify if inhibition of *Shigella* infection upon stress induced by arsenite is a general phenomenon. We have validated this effect in additional cell lines also used as models/targets for *Shigella* infection studies, specifically HT-29 and Caco-2 cells. In both cell lines, we observed a significant reduction of *Shigella* infection upon arsenite treatment (Fig. EV1C and EV1D of the revised manuscript).

The reason for the different extent of the arsenite inhibitory effect in HeLa versus HCT-8 cells is currently unclear. We initially hypothesized that this might be related to different levels of p38 induction in the two cell types, which would likely lead to differences in ASM activation and ceramide production. However, we have not detected significant differences between the two cell types in terms of p38 activation (Western blot below).

It is conceivable that membrane remodeling upon arsenite is more efficient in HeLa cells than HCT-8, despite similar extents of p38 activation. However there might be other contributing factors, such as basal plasma membrane lipid composition, membrane proteins or membrane curvature that may explain the observed differences. Nonetheless, the inhibitory effect of the stressors was consistent in all the epithelial cells tested.

2) It is unclear why inhibition of protein synthesis by puromycin and cycloheximide did not affect *Shigella* infection (Fig. S1C-E), but inhibition of protein synthesis by anisomycin (Fig. 1E-F) did? Further, the effect of stress induced by anisomycin on *Shigella* infection was overcome by inhibiting ASM activity. However, how does the cell make ASM in the first place in the presence of a protein synthesis inhibitor? Is ASM present (but inactive?) before treatment with anisomycin? This would contradict with the results shown in Fig. S4 i.e. that the activation of the p38 MAPK pathway by arsenite and anisomycin is required for ASM activation.

Reply: The reason underlying the inhibition of *Shigella* infection by anisomycin is an important point and was clarified in the revised version of the manuscript. As the Referee indicates, anisomycin inhibits

protein synthesis, but this compound also leads to ROS production [e.g. (Chambers & LoGrasso, 2011; Torocsik & Szeberenyi, 2000)] and to p38 MAPK activation (Fig. 4A). The inhibitory effect of anisomycin on *Shigella* is indeed unrelated to its effect on protein synthesis and primarily due to the effect of anisomycin in inducing oxidative stress. It should also be noted that the treatment duration is short (20 min), and therefore should not impact the expression levels of most proteins, given that in human cells the typical half-life of proteins is significantly longer (Eden et al, 2011).

Nonetheless, regarding ASM production in the presence of a protein synthesis inhibitor it should be noted that the protein does not need to be produced de novo to be activated. In fact, ASM is usually present inside cells associated with the lysosomal compartment; upon activation, there is a redistribution of the protein to the outer leaflet of the plasma membrane, where it hydrolyzes sphingomyelin giving rise to ceramide-rich platforms [(Grassme et al, 2001) and reviewed in (Beckmann et al, 2014)].

Regarding the results shown in Fig. S4 (Fig. EV3 of the revised manuscript), we do not fully understand the comment of the Referee. In the presence of anisomycin, since the treatment duration is relatively short (20 min), no changes in total p38 levels are detected – activation of p38 occurs instead by phosphorylation. This is evident in Fig. 4A and EV3G and in agreement with the previously published observations that oxidative stress leads to activation (by phosphorylation) of p38 MAPK pathway.

3) The effect of cellular stress induction on *Salmonella* infection is inconsistent. The authors state that upon infection with *Salmonella*, fewer cells were infected, but more bacteria adhered per cell (based on fluorescence microscopy analyses; Fig. 4A, D-F).

However, the number of intracellular *Salmonella* (based on CFU counts and *Salmonella* mRNA quantification) were not affected (Fig. 4B-C)?

Would it be possible that *Salmonella* replicates more efficiently inside arsenite/anisomycin-stressed cells?

Reply: As the Referee notes, we have indeed observed that the pattern of *Salmonella* infection is different in cells treated with the stress inducers, showing fewer infected cells, but more bacteria per infected cell (Fig. 5A, 5D-F, S3A). As such, the higher number of bound bacteria per cell compensates for the lower number of infected cells, translating into a total number of CFU counts and *Salmonella* mRNA levels (Fig. 5B, 5C) that are comparable in cells treated with the stress inducers and in control cells.

Concerning the possibility that *Salmonella* could replicate more efficiently inside arsenite- and anisomycin-treated cells, it should be noted that in these experiments *Salmonella* infection was analyzed at early times post-infection, specifically 0.5 hpi (Fig. 5A) and 10 min pi (Fig. 5D-F, S3A), time-points at which *Salmonella* replication is not yet occurring; *Salmonella* replication within HeLa cells typically starts at around 4-6 hpi.

To evaluate a possible effect of arsenite on *Salmonella* replication, we have performed additional experiments in which cells were treated with arsenite only after *Salmonella* invasion (at 0.5 hpi for 1 h), thus excluding the effect of the stressors on *Salmonella* binding (more bacteria per infected cell, albeit less infected cells). We have then evaluated the pattern of infection by microscopy and quantified the total CFUs at 20 hpi, a time-point at which *Salmonella* is in full-blown replication within host cells. The results from these experiments clearly show that *Salmonella* replication is not affected by arsenite treatment (Fig. S3B and S3C of the revised manuscript).

4) The authors demonstrate that the infection with *Shigella* induces ASM activation and ceramide accumulation also in bystander cells.

This suggests a general epithelial cell defense mechanism, where bystander cells next to infected cells would be immune to infection, i.e. ceramide-positive cells should not be infected by *Shigella*, which might be easily tested using e.g. flow cytometry analysis.

Reply: The Referee's thoughts on the role and response of bystander cells during infection are very interesting, alas difficult to follow-up within the timeframe of a manuscript revision. In fact, this is a topic in which we are very interested in and will pursue in the future.

Regarding the approach suggested by the Referee of applying flow-cytometry to confirm the microscopy-based observation that bystander cells have high levels of ceramide, it appears simple but it revealed to be rather cumbersome due to technical reasons. As shown in the microscopy images in the original manuscript (Fig. 7B, 7G, 7H, EV5P) and low magnification images introduced in the revised version (Fig. EV5Q), the increase of ceramide occurs essentially in cells in the close vicinity to cells with high number of intracellular *Shigella*. Given that it is impossible to select the bystander cells in close proximity to *Shigella* infected cells by flow-cytometry, the analysis is necessarily biased by the bystander cells distant from infected cells. Therefore, we reasoned that this approach would not be conclusive. We have nonetheless performed these experiments in order to quantify the relative ceramide production in *Shigella* infected cells compared to mock-treated cells (see Referee 2, points 4 and 6). An increase of the mean fluorescence intensity of the ceramide signal was detected in *Shigella* infected cells compared to control cells (total population, Fig. 7D of the revised manuscript). For these experiments, live cells were labeled with the ceramide antibody followed by labelling with fluorescently labelled secondary antibody. Interestingly, in the bystander cells the ceramide signal is higher than that of mock-treated cells.

Minor comments:

1) The fluorescence images in Figs 1B, 1C, 1E, 2A, 2D etc. would benefit from quantification. Compared to the shown CFU data, the impaired *Shigella* infection is less convincing in the presented fluorescence images.

Reply: As suggested by the Referee, we have performed the quantification of fluorescence microscopy images corresponding to key experiments shown in Figs 1 and 2, using an automated image analysis approach we have previously reported (Maudet et al, 2014; Sunkavalli et al, 2017). The quantification of the fluorescence microscopy images (Fig. EV1A and EV1H) agrees with the CFU data. It should be noted, however, that these two assays quantify different parameters: the CFU reveals total bacterial loads while the microscopy shows differences in the percentage of infected cells (independently of the intracellular load). As such, the fold changes in terms of percentage of infected cells are expected to be lower than for total bacterial counts.

2) The swimming behavior of *Salmonella* near the surface of host cells has been extensively studied before and should be cited (PMID 22911370) - page 8, top.

Reply: We thank the Referee for calling our attention to this missing citation and we have corrected the manuscript accordingly.

3) The use of a $\Delta fliC$ *Salmonella* mutant as a non-motile strain is curious. As noted by the authors, *Salmonella* harbors two distinct flagellins (FliC and FljB), whose expression is stochastic. Thus, a subset of a population of $\Delta fliC$ mutant *Salmonella* would still be motile (which is also noted by the authors). However, the frequency of switching flagellin expression is low (around 1/5000 per cell per generation) and biased towards switching to the *fliC*_{ON} orientation (PMID 1848842). Accordingly, a population of $\Delta fliC$ mutant *Salmonella* would primarily be non-motile and the infection phenotype similar to a complete flagella mutant ($\Delta flhC$), which is consistent with the author's results shown in Fig. 4G-H. This should be noted in the manuscript and the rationale of using both a $\Delta fliC$ and $\Delta flhC$ mutant should be explained.

Reply: As recommended by the Referee, we have discussed in more detail the differences between the *Salmonella* $\Delta fliC$ and $\Delta flhC$ mutant strains in the revised version of the manuscript, describing that the majority of the $\Delta fliC$ mutant population is non-motile. We have used both strains in order to validate the results using two essentially non-motile strains and also because the $\Delta fliC$ /pFliC complemented strain used as control was already available in the laboratory. These results are further supported by experiments with a *Salmonella* $\Delta motA$ mutant (see point 5 of Referee 3).

4) Is the ability of *Salmonella* to accumulate at the remaining permissive entry sites dependent on the expressed flagellin? It has been shown that the near-surface swimming behavior of *Salmonella* is dependent on which flagellin (FliC or FljB) is expressed (PMID 28295924), which might fit with the author's observations.

Reply: As suggested by the Referee we have evaluated if the ability of *Salmonella* to accumulate in the remaining permissive sites in host cells exposed to stress is dependent on which flagellin is expressed. This is an interesting hypothesis, given that it has been recently shown by Marc Erhardt's group that *Salmonella* expressing FliC-flagella does better at identifying target sites on host cell surface and invasion (Horstmann et al, 2017). For this purpose, we requested the SL1344 $\Delta hin-5717::FRT$ (*fliC*^{ON}) and SL1344 $\Delta hin-5718::FRT$ (*fliB*^{ON}) strains from the Erhardt lab; these strains are locked in either FliC- or FljB expression. However, when evaluating the effect of arsenite on infection by these strains we did not observe a difference in binding to host cells. This is most likely a consequence of the use of centrifugation to promote/enforce the contact of the bacteria with the epithelial cell surface [see below and in agreement with results reported by the Erhardt lab in (Horstmann et al, 2017)]. In terms of the effect of stress, arsenite pre-treatment slightly affected the binding of the *fliC*^{ON} and *fliB*^{ON} strains to host cells, but the extent of inhibition was lower than that of the non-motile strains (Fig. 5H). Based on these data, we conclude that, in the used experimental conditions, the ability of *Salmonella* to accumulate in the remaining permissive entry sites upon stress is not dependent in the expressed flagellin.

5) Another interesting experiment would be to mix a population of non-motile and motile *Salmonella* bacteria (labelled with different fluorescence proteins or different antibiotic resistance markers) and perform a co-infection experiment. The author's model would predict that only motile bacteria would be able to infect and thus out-compete a non-motile *Salmonella* mutant. Such an experiment would nicely complement the author's *Shigella-Salmonella* co-infection experiment while circumventing any inter-species effects.

In this respect, I would recommend to use a *Salmonella* mutant of the motor-force generators (MotAB), which impairs rotation (and thus motility function) of the flagellum, but does not affect the role of flagella in binding to surfaces.

Reply: We fully agree with the Referee that this is an interesting experiment that complements the *Shigella-Salmonella* re-infection experiments. As suggested by the Referee, we have performed co-infection experiments with *Salmonella* WT and $\Delta motA$ mutant strain. As expected, similarly to the *Salmonella* $\Delta fliC$ and $\Delta fliH$ mutant strains, the infection with *Salmonella* $\Delta motA$ alone was inhibited by arsenite pre-treatment (see below). In the co-infection experiment (MOI 25 of each bacterial strain, i.e. total MOI 50), we observed that arsenite pre-treatment leads to a significant decrease of $\Delta motA$ infection, whereas the *Salmonella* WT infection is not affected (see below). However, a less pronounced inhibitory effect of arsenite in the co-infection experiments was observed, when compared to the $\Delta motA$ infection alone. This is likely explained by the favored infection of the $\Delta motA$ strain due to cooperative invasion with the WT *Salmonella*. This is in line with previous observation of cooperative invasion of non-motile strains or non-invasive strains with WT *Salmonella* [e.g. (Lorkowski et al, 2014; Misselwitz et al, 2012)].

6) Does a primary infection with *Salmonella* induce host membrane remodeling and prevent a secondary infection with *Shigella*?

Reply: As described in point 2 of Referee 1, we did not observe ceramide increase in response to *Salmonella* infection/replication (Fig. S3F), indicating that membrane remodeling does not occur during *Salmonella* infection. In agreement, re-infection experiments showed that *Salmonella* primary infection (18 h) did not affect *Shigella* secondary infection (Fig EV4M and EV4N of the revised manuscript).

7) The bacterial gene nomenclature throughout the manuscript should be verified (i.e. the correct nomenclature of gene deletions is lowercase and italics, e.g. $\Delta fliC$)

Reply: We have corrected the revised version of the manuscript accordingly.

Additional suggestions:

1) Fig. 5C; 5E: How were *Shigella* colonies arising from the primary and secondary infection determined (also, how were *Shigella* and *Salmonella* colonies discriminated)?

Did the authors use different antibiotic resistance cassettes? If so, this should be mentioned in the Methods section.

Reply: Discrimination of the colonies derived from primary and secondary infections was possible since the strains used for the re-infection experiments have indeed different antibiotic resistance cassettes, as follows: *Shigella*-mCherry, kanamycin resistance and *Salmonella*-mCherry, ampicillin resistance (primary infection); *Shigella*-GFP and *Salmonella*-GFP, chloramphenicol resistance (secondary infections). This information has been introduced in the Materials and Methods section of the revised manuscript.

2) Discussion: Temperature-sensitive flagella expression of *Listeria* and *Yersinia*. My general understanding is that during the initial colonization of the gut, the pathogens are still motile, since infection of the epithelium occurs rapidly after ingestion of the pathogens (which expressed flagella outside the host).

The temperature-dependent downregulation of flagella expression would only affect de novo synthesis of flagella and might be more important for intracellular survival and systemic infection?

Reply: We have introduced data in the revised manuscript showing that infection by *Listeria* and *Yersinia*, grown at 37C, is inhibited by arsenite treatment (Fig 5I-L of the revised manuscript). The rationale behind introducing these results in the manuscript is to further highlight that stress induced membrane remodeling protects epithelial cells from infection by non-motile bacterial pathogens. We fully agree with the Referee that the temperature-dependent downregulation of flagella expression of *Listeria* and *Yersinia* would most likely affect the systemic infection, rather than the initial colonization of gut epithelial cells.

3) Page 11: rephrase '...leading to inhibition *Salmonella* invasion...'

Reply: We have corrected the sentence to 'leading to inhibition of *Salmonella* invasion'.

REFERENCES

Arbibe L, Kim DW, Batsche E, Pedron T, Mateescu B, Muchardt C, Parsot C, Sansonetti PJ (2007) An injected bacterial effector targets chromatin access for transcription factor NF-kappaB to alter transcription of host genes involved in immune responses. *Nature immunology* **8**: 47-56

Awojoodu AO, Keegan PM, Lane AR, Zhang Y, Lynch KR, Platt MO, Botchwey EA (2014) Acid sphingomyelinase is activated in sickle cell erythrocytes and contributes to inflammatory microparticle generation in SCD. *Blood* **124**: 1941-1950

Bagam P, Singh DP, Inda ME, Batra S (2017) Unraveling the role of membrane microdomains during microbial infections. *Cell biology and toxicology*

Beckmann N, Sharma D, Gulbins E, Becker KA, Edelmann B (2014) Inhibition of acid sphingomyelinase by tricyclic antidepressants and analogs. *Frontiers in physiology* **5**: 331

Beuzon CR, Salcedo SP, Holden DW (2002) Growth and killing of a *Salmonella enterica* serovar Typhimurium sifA mutant strain in the cytosol of different host cell lines. *Microbiology* **148**: 2705-2715

Bianco F, Perrotta C, Novellino L, Francolini M, Riganti L, Menna E, Saggiotti L, Schuchman EH, Furlan R, Clementi E, Matteoli M, Verderio C (2009) Acid sphingomyelinase activity triggers microparticle release from glial cells. *The EMBO journal* **28**: 1043-1054

Chambers JW, LoGrasso PV (2011) Mitochondrial c-Jun N-terminal kinase (JNK) signaling initiates physiological changes resulting in amplification of reactive oxygen species generation. *The Journal of biological chemistry* **286**: 16052-16062

Cumming G, Fidler F, Vaux DL (2007) Error bars in experimental biology. *The Journal of cell biology* **177**: 7-11

Edelmann B, Bertsch U, Tchikov V, Winoto-Morbach S, Perrotta C, Jakob M, Adam-Klages S, Kabelitz D, Schütze S (2011) Caspase-8 and caspase-7 sequentially mediate proteolytic activation of acid sphingomyelinase in TNF-R1 receptosomes. *The EMBO journal* **30**: 379-394

Eden E, Geva-Zatorsky N, Issaeva I, Cohen A, Dekel E, Danon T, Cohen L, Mayo A, Alon U (2011) Proteome half-life dynamics in living human cells. *Science* **331**: 764-768

Faulstich M, Hagen F, Avota E, Kozjak-Pavlovic V, Winkler AC, Xian Y, Schneider-Schaulies S, Rudel T (2015) Neutral sphingomyelinase 2 is a key factor for PorB-dependent invasion of *Neisseria gonorrhoeae*. *Cellular microbiology* **17**: 241-253

Gorelik A, Illes K, Heinz LX, Superti-Furga G, Nagar B (2016) Crystal structure of mammalian acid sphingomyelinase. *Nature communications* **7**: 12196

Grassme H, Jekle A, Riehle A, Schwarz H, Berger J, Sandhoff K, Kolesnick R, Gulbins E (2001) CD95 signaling via ceramide-rich membrane rafts. *The Journal of biological chemistry* **276**: 20589-20596

Grassme H, Jendrossek V, Riehle A, von Kurthy G, Berger J, Schwarz H, Weller M, Kolesnick R, Gulbins E (2003) Host defense against *Pseudomonas aeruginosa* requires ceramide-rich membrane rafts. *Nature medicine* **9**: 322-330

Grassme H, Riehle A, Wilker B, Gulbins E (2005) Rhinoviruses infect human epithelial cells via ceramide-enriched membrane platforms. *The Journal of biological chemistry* **280**: 26256-26262

Grimm MO, Grimm HS, Patzold AJ, Zinser EG, Halonen R, Duering M, Tschape JA, De Strooper B, Müller U, Shen J, Hartmann T (2005) Regulation of cholesterol and sphingomyelin metabolism by amyloid-beta and presenilin. *Nature cell biology* **7**: 1118-1123

Horstmann JA, Zschieschang E, Truschel T, de Diego J, Lunelli M, Rohde M, May T, Strowig T, Stradal T, Kolbe M, Erhardt M (2017) Flagellin phase-dependent swimming on epithelial cell surfaces contributes to productive *Salmonella* gut colonisation. *Cellular microbiology* **19**

Kasper CA, Sorg I, Schmutz C, Tschon T, Wischnewski H, Kim ML, Arrieumerlou C (2010) Cell-cell propagation of NF-kappaB transcription factor and MAP kinase activation amplifies innate immunity against bacterial infection. *Immunity* **33**: 804-816

King TJ, Fukushima LH, Donlon TA, Hieber AD, Shimabukuro KA, Bertram JS (2000) Correlation between growth control, neoplastic potential and endogenous connexin43 expression in HeLa cell lines: implications for tumor progression. *Carcinogenesis* **21**: 311-315

Kumagai Y, Sumi D (2007) Arsenic: signal transduction, transcription factor, and biotransformation involved in cellular response and toxicity. *Annual review of pharmacology and toxicology* **47**: 243-262

Lafont F, van der Goot FG (2005) Bacterial invasion via lipid rafts. *Cellular microbiology* **7**: 613-620

Li H, Xu H, Zhou Y, Zhang J, Long C, Li S, Chen S, Zhou JM, Shao F (2007) The phosphothreonine lyase activity of a bacterial type III effector family. *Science* **315**: 1000-1003

Liu SX, Athar M, Lippai I, Waldren C, Hei TK (2001) Induction of oxyradicals by arsenic: implication for mechanism of genotoxicity. *Proceedings of the National Academy of Sciences of the United States of America* **98**: 1643-1648

Lorkowski M, Felipe-Lopez A, Danzer CA, Hansmeier N, Hensel M (2014) *Salmonella enterica* invasion of polarized epithelial cells is a highly cooperative effort. *Infection and immunity* **82**: 2657-2667

Lu DY, Chen HC, Yang MS, Hsu YM, Lin HJ, Tang CH, Lee CH, Lai CK, Lin CJ, Shyu WC, Lin FY, Lai CH (2012) Ceramide and Toll-like receptor 4 are mobilized into membrane rafts in response to *Helicobacter pylori* infection in gastric epithelial cells. *Infection and immunity* **80**: 1823-1833

Luberto C, Hassler DF, Signorelli P, Okamoto Y, Sawai H, Boros E, Hazen-Martin DJ, Obeid LM, Hannun YA, Smith GK (2002) Inhibition of tumor necrosis factor-induced cell death in MCF7 by a novel inhibitor of neutral sphingomyelinase. *The Journal of biological chemistry* **277**: 41128-41139

Luisoni S, Suomalainen M, Boucke K, Tanner LB, Wenk MR, Guan XL, Grzybek M, Coskun U, Greber UF (2015) Co-option of Membrane Wounding Enables Virus Penetration into Cells. *Cell host & microbe* **18**: 75-85

Malik-Kale P, Winfree S, Steele-Mortimer O (2012) The bimodal lifestyle of intracellular *Salmonella* in epithelial cells: replication in the cytosol obscures defects in vacuolar replication. *PLoS one* **7**: e38732

Maudet C, Mano M, Sunkavalli U, Sharan M, Giacca M, Forstner KU, Eulalio A (2014) Functional high-throughput screening identifies the miR-15 microRNA family as cellular restriction factors for *Salmonella* infection. *Nature communications* **5**: 4718

Misselwitz B, Barrett N, Kreibich S, Vonaesch P, Andrichke D, Rout S, Weidner K, Sormaz M, Songhet P, Horvath P, Chabria M, Vogel V, Spori DM, Jenny P, Hardt WD (2012) Near surface swimming of *Salmonella Typhimurium* explains target-site selection and cooperative invasion. *PLoS pathogens* **8**: e1002810

Petersen NH, Olsen OD, Groth-Pedersen L, Ellegaard AM, Bilgin M, Redmer S, Ostefeld MS, Ulanet D, Dovmark TH, Lonborg A, Vindelov SD, Hanahan D, Arenz C, Ejsing CS, Kirkegaard T, Rohde M, Nylandsted J, Jaattela M (2013) Transformation-associated changes in sphingolipid metabolism sensitize cells to lysosomal cell death induced by inhibitors of acid sphingomyelinase. *Cancer cell* **24**: 379-393

Pralle A, Keller P, Florin EL, Simons K, Horber JK (2000) Sphingolipid-cholesterol rafts diffuse as small entities in the plasma membrane of mammalian cells. *The Journal of cell biology* **148**: 997-1008

Ruiz-Ramos R, Lopez-Carrillo L, Rios-Perez AD, De Vizcaya-Ruiz A, Cebrian ME (2009) Sodium arsenite induces ROS generation, DNA oxidative damage, HO-1 and c-Myc proteins, NF-kappaB activation and cell proliferation in human breast cancer MCF-7 cells. *Mutation research* **674**: 109-115

Schieven GL (2005) The biology of p38 kinase: a central role in inflammation. *Current topics in medicinal chemistry* **5**: 921-928

Sezgin E, Levental I, Mayor S, Eggeling C (2017) The mystery of membrane organization: composition, regulation and roles of lipid rafts. *Nature reviews Molecular cell biology* **18**: 361-374

Simonis A, Hebling S, Gulbins E, Schneider-Schaulies S, Schubert-Unkmeir A (2014) Differential activation of acid sphingomyelinase and ceramide release determines invasiveness of *Neisseria meningitidis* into brain endothelial cells. *PLoS pathogens* **10**: e1004160

Stecher B, Barthel M, Schlumberger MC, Haberli L, Rabsch W, Kremer M, Hardt WD (2008) Motility allows *S. Typhimurium* to benefit from the mucosal defence. *Cellular microbiology* **10**: 1166-1180

Stecher B, Hapfelmeier S, Muller C, Kremer M, Stallmach T, Hardt WD (2004) Flagella and chemotaxis are required for efficient induction of *Salmonella enterica* serovar Typhimurium colitis in streptomycin-pretreated mice. *Infection and immunity* **72**: 4138-4150

Sullivan LM, Weinberg J, Keane JF, Jr. (2016) Common Statistical Pitfalls in Basic Science Research. *Journal of the American Heart Association* **5**

Sunkavalli U, Aguilar C, Silva RJ, Sharan M, Cruz AR, Tawk C, Maudet C, Mano M, Eulalio A (2017) Analysis of host microRNA function uncovers a role for miR-29b-2-5p in *Shigella* capture by filopodia. *PLoS pathogens* **13**: e1006327

Tonnetti L, Veri MC, Bonvini E, D'Adamio L (1999) A role for neutral sphingomyelinase-mediated ceramide production in T cell receptor-induced apoptosis and mitogen-activated protein kinase-mediated signal transduction. *The Journal of experimental medicine* **189**: 1581-1589

Torocsik B, Szeberenyi J (2000) Anisomycin uses multiple mechanisms to stimulate mitogen-activated protein kinases and gene expression and to inhibit neuronal differentiation in PC12 pheochromocytoma cells. *The European journal of neuroscience* **12**: 527-532

Varma R, Mayor S (1998) GPI-anchored proteins are organized in submicron domains at the cell surface. *Nature* **394**: 798-801

Vassilieva EV, Gerner-Smidt K, Ivanov AI, Nusrat A (2008) Lipid rafts mediate internalization of beta1-integrin in migrating intestinal epithelial cells. *American journal of physiology Gastrointestinal and liver physiology* **295**: G965-976

Yang Y, Kim SC, Yu T, Yi YS, Rhee MH, Sung GH, Yoo BC, Cho JY (2014) Functional roles of p38 mitogen-activated protein kinase in macrophage-mediated inflammatory responses. *Mediators of inflammation* **2014**: 352371

Thanks for submitting your revised version to The EMBO journal.

Your study has now been the original referees and their comments are provided below. The referees appreciate the included revisions and find that the manuscript has been strengthened. Referee #2 still has a number of concerns that I would like to ask you to address in a revised version. In particular you have to provide further data to sort out the specificity of the ASM antibody. A better presentation of the data concerning the enzyme activity is also needed. It would be good to show enzyme units and not percentage, and clarify how you did the data analysis. Not clear how the control value was set at 100% is that the mean of all independent control samples? Please clarify how the data analysis was done. I also agree with referee #3 in that it would be good to show the graphs as dot plots where you depict all the individual data points. You can also provide all the original data points as source data as well.

 REFEREE REPORTS:

Referee #1:

This significantly revised manuscript is clever and convincing, and shows a new role for stress induced host membrane remodeling in host defense. This article impacts several cutting edge topics in infection biology (including inflammation, bystander cell activation, re-infection, commensal bacteria), and would benefit from a News & Views to bring to the attention of a wider community.

I have some comments, mostly of interest for future investigation.

1. Testing the whole animal impact of this discovery is complicated to envision, moreover work performed in vivo using the guinea pig are complicated. Different systems or animal models may be required.
2. Both ASM and ceramide accumulate in bystander cells. The authors now have a wonderful system to investigate the underlying mechanism and consequence of bystander cell biology, helpful to support the few studies currently available on this subject.
3. How important is membrane remodeling to the host response to infection / inflammation? Links proposed by the authors that this mechanism might be relevant to restrict commensal barrier translocation will also be exciting to follow up.
4. Figures are clear and of high quality.
 - Small typo bottom page 10: "whereas infection with the Δ fliC/pFilC" ... should be pFliC

Referee #2:

The authors addressed some of my point. However, several issues remain:

1. Rafts have been shown to be very small and it is almost impossible to visualize them. Membrane domains formed by ceramide are large, as nicely shown in many published studies. However, the figures do not show ceramide domains. The figures show rather small domains. The flow cytometry data are nice, but the identification of membrane domains as described in the manuscript still requires definition.

Amitriptyline also blocks the acid ceramidase and genetic data are required to support the notion. The same applies to GW4869, which is rather toxic. Genetic models are available and must be used. However, at least siRNA data are provided.

The activation of the acid sphingomyelinase is still insufficiently reported. The data show an increase from 100% to 120%. The variation in the treated samples is high. The units in the legend

are not enzyme units and the variation is not given (citation: the ASM activity in mock treated cells corresponds to 0.50 mU/h). What are mU? Enzyme units? Original data are not provided. The controls are set at 100% and, thus, have no variation. It remains unclear how the statistics were performed (using the original data or the data set at 100%). However, I have clearly asked to give the absolute activity of the acid sphingomyelinase and it remains unclear why the authors to refuse to show the original data.

It seems extremely unlikely that all activities are 0.48 mU/h (whatever the protein concentration is) or 50 mU/h.

As given, the small increase could be also due to different cell numbers.

This is very important, because the authors claim an activation of the acid sphingomyelinase, which remains unproven.

The western blots on the acid sphingomyelinase show many bands and they are almost all reduced to the controls. How can that be? Are all of these proteins acid sphingomyelinase isoforms, even higher than 75 kDa? It suggests that the antibody is simply unspecific and the blots are not meaningful.

Referee #3:

Tawk et al. have extensively revised the manuscript and appropriately addressed all my concerns. The additional experiments nicely support the author's conclusions and I can only congratulate the authors for this exciting work.

Minor comment:

In respect to comment #3 of referee 2. Perhaps it would be useful to represent the data as dot plots showing the individual data points rather than as bar graphs.

'Stress-induced host membrane remodeling protects from infection by non-motile bacterial pathogens'

Referee #1:

This significantly revised manuscript is clever and convincing, and shows a new role for stress induced host membrane remodeling in host defense. This article impacts several cutting edge topics in infection biology (including inflammation, bystander cell activation, re-infection, commensal bacteria), and would benefit from a News & Views to bring to the attention of a wider community.

I have some comments, mostly of interest for future investigation.

1. Testing the whole animal impact of this discovery is complicated to envision, moreover work performed in vivo using the guinea pig are complicated. Different systems or animal models may be required.
2. Both ASM and ceramide accumulate in bystander cells. The authors now have a wonderful system to investigate the underlying mechanism and consequence of bystander cell biology, helpful to support the few studies currently available on this subject.
3. How important is membrane remodeling to the host response to infection / inflammation? Links proposed by the authors that this mechanism might be relevant to restrict commensal barrier translocation will also be exciting to follow up.
4. Figures are clear and of high quality.

Reply: We are very grateful to the Referee for her/his enthusiastic and supportive comments. We fully agree that this constitutes a very good system to study bystander cell biology in the context of infection biology and that further work is necessary to clarify how widespread is the relevance of membrane remodeling in host-microbial interactions. These are research lines that we will pursue in future studies.

- Small typo bottom page 10: "whereas infection with the Δ fliC/pFilC" ... should be pFliC

Reply: We have corrected this typo in the revised version of the manuscript.

Referee #2:

The authors addressed some of my point. However, several issues remain:

Reply: We would like to thank the Referee for her/his comments that allowed us to clarify some additional points in the manuscript, most importantly regarding the measurement of the acid sphingomyelinase activity.

1. Rafts have been shown to be very small and it is almost impossible to visualize them. Membrane domains formed by ceramide are large, as nicely shown in many published studies. However, the figures

do not show ceramide domains. The figures show rather small domains. The flow cytometry data are nice, but the identification of membrane domains as described in the manuscript still requires definition.

Reply: We agree with the Referee that it will be very interesting to further characterize the ceramide domains, but we think this is outside the scope of this manuscript. In the future, we plan to apply super-resolution and biophysics techniques to further characterize these platforms.

Amitriptyline also blocks the acid ceramidase and genetic data are required to support the notion. The same applies to GW4869, which is rather toxic. Genetic models are available and must be used. However, at least siRNA data are provided.

Reply: Given that we have demonstrated similar phenotypes with amitriptyline and the ASM knockdown by siRNA (Figures 3A-I, 6I-K, EV5A-C), we have excluded that the amitriptyline phenotype is related to acid ceramidase inhibition.

The activation of the acid sphingomyelinase is still insufficiently reported. The data show an increase from 100% to 120%. The variation in the treated samples is high. The units in the legend are not enzyme units and the variation is not given (citation: the ASM activity in mock treated cells corresponds to 0.50 mU/h). What are mU? Enzyme units? Original data are not provided. The controls are set at 100% and, thus, have no variation. It remains unclear how the statistics were performed (using the original data or the data set at 100%). However, I have clearly asked to give the absolute activity of the acid sphingomyelinase and it remains unclear why the authors to refuse to show the original data.

It seems extremely unlikely that all activities are 0.48 mU/h (whatever the protein concentration is) or 50 mU/h.

As given, the small increase could be also due to different cell numbers.

This is very important, because the authors claim an activation of the acid sphingomyelinase, which remains unproven.

Reply: According to the Referee's comment, we have provided in the revised manuscript a more exhaustive description of the methodology used for the measurement and calculation of the acid sphingomyelinase activity ('Plasma membrane acid sphingomyelinase activity assays section' of 'Materials and Methods'). Moreover, we have now provided the data as ASM enzymatic activity, rather than presenting the data normalized to the control (Figures 7C, EV2K and EV2P), as requested by the Referee.

For these experiments, we have prepared membrane fractions (as described previously by Tonnetti et al, 1999). Importantly, 1.2×10^6 cells were used as input for the extraction of the membrane fractions, for each of the independent experiments and conditions, and therefore the changes in activity cannot be attributed to different cell numbers. 10 μ l of the membrane fractions (i.e. corresponding to 3.0×10^5 cells per condition) were used for the measurement of acid sphingomyelinase activity, which was performed using the Amplex Red Sphingomyelinase Assay Kit (Invitrogen, A12220). For the new set of experiments provided in the revised manuscript a standard curve was established for each experiment, based on 1:4 serial dilutions of the commercial sphingomyelinase provided in the Amplex kit (10U/ml; 1U corresponds

to the amount of enzyme required to hydrolyze 1 μ mole substrate in 1 min at 37°C), starting from 4×10^{-3} U to 0.1×10^{-5} U (these values were incorrectly stated in the previous version of the manuscript by a 10x dilution factor). The reactions (standard curve and samples) were incubated at 37°C for 1 hour and fluorescence was measured with a microplate reader using excitation at 530 nm and emission at 590 nm. For each experiment, the fluorescence of the samples was converted to enzyme units using a standard curve.

The western blots on the acid sphingomyelinase show many bands and they are almost all reduced to the controls. How can that be? Are all of these proteins acid sphingomyelinase isoforms, even higher than 75 kDa? It suggests that the antibody is simply unspecific and the blots are not meaningful.

Although we understand the Reviewer point regarding the Western-blot, we have provided evidence in the manuscript that the antibody works very well in immunofluorescence (dramatic decrease of signal upon ASM knockdown, demonstrating the specificity of the antibody staining; Figure S2B). Indeed, immunofluorescence is the main application of the antibody throughout the manuscript, to demonstrate ASM relocalization. We have used this antibody in Western-blotting only to confirm the efficiency of ASM knockdown, which we have validated by additional approaches, namely immunofluorescence, qRT-PCR and ASM enzymatic activity assays (Figures S2B, EV2N and EV2P). Nonetheless, we have optimized the conditions for the use of the antibody for Western-blotting (dilution – 1:200; blocking conditions – TSB with 2% milk and 3% BSA). In these conditions, only 3 bands appear in the blot (Figure EV2O and S4), namely a protein of 55-70 kDa (corresponding to the mature ASM), a protein of 70-100 kDa (presumably correspond to the proenzyme) and a lower molecular weight protein (35-55 kDa) that to our knowledge has not been described before. Importantly, the levels of all 3 proteins are reduced in ASM siRNA transfected cells. Thus, we think that the results of the blot further corroborate the reduction of ASM levels upon siRNA treatment.

Referee #3:

Tawk et al. have extensively revised the manuscript and appropriately addressed all my concerns. The additional experiments nicely support the author's conclusions and I can only congratulate the authors for this exciting work.

We are grateful to the Referee for her/his very positive evaluation of the revised version of the manuscript.

Minor comment:

In respect to comment #3 of referee 2. Perhaps it would be useful to represent the data as dot plots showing the individual data points rather than as bar graphs.

As suggested by the Referee we have introduced in the revised version of the manuscript the individual data points for each graph, albeit keeping the bar graphs, for easier visualization of the results.

Thanks for sending us your revised manuscript. I asked referee #3 to take a look at the revised version and I have now received the feedback. As you can see below the referee appreciates the introduced changes. I am therefore very pleased to accept the manuscript for publication here.

Before sending you the formal acceptance letter there are just a few issues to sort out.

- There is a callout to S6E-J in figure legends for EV4. Please fix that.
- Regarding Figure 7G - please upload a modified figure showing the white background in the upper left corners. Please also make sure that you mention in figure legends or methods how the images were acquired and assembled.
- Our publisher is doing a pre-publication check on the manuscript and I am still waiting for their comments, but should hopefully receive them later today. I will send you the document as soon as I receive it so that you incorporate their suggestions. Please wait to submit the revised manuscript until their suggestions have been incorporated.

That should be all - as soon as I get the revised version in I will send you the formal acceptance letter.

REFeree REPORTS:

Referee #3:

The authors appropriately addressed the remaining issues in the revised manuscript. The improved method to measure acid sphingomyelinase is convincing.

Corresponding Author Name: Ana Eulalio

Manuscript Number: EMBOJ-2017-98529